# Covariate-moderated Empirical Bayes Matrix Factorization

**William R. P. Denault, Karl Tayeb, Peter Carbonetto & Matthew Stephens**
Departments of Statistics and Human Genetics
University of Chicago
Chicago, IL 60637, USA
{wdenault,ktayeb,pcarbo,mstephens}@uchicago.edu

**Jason Willwerscheid**
Mathematics and Computer Science
Providence College
Providence, RI 02918, USA
jwillwer@providence.edu

## Abstract

Matrix factorization is a fundamental method in statistics and machine learning for inferring and summarizing structure in multivariate data. Modern data sets often come with "side information" of various forms (images, text, graphs) that can be leveraged to improve estimation of the underlying structure. However, existing methods that leverage side information are limited in the types of data they can incorporate, and they assume specific parametric models. Here, we introduce a novel method for this problem, *covariate-moderated empirical Bayes matrix factorization* (cEBMF). cEBMF is a modular framework that accepts any type of side information that is processable by a probabilistic model or a neural network. The cEBMF framework can accommodate different assumptions and constraints on the factors through the use of different priors, and it adapts these priors to the data. We demonstrate the benefits of cEBMF in simulations and in analyses of spatial transcriptomics and collaborative filtering data. A PyTorch-based implementation of cEBMF with flexible priors is available at https://github.com/william-denault/cebmf_torch.

## 1 Introduction

Matrix factorization methods, which include principal component analysis (PCA), factor analysis, and non-negative matrix factorization (NMF) [1–3], are very widely used methods for inferring latent structure from data, performing exploratory data analyses, and visualizing large data sets (e.g., [4–6]). Matrix factorization methods are also instrumental in other statistical analyses such as adjusting for unobserved confounding [7]. When factorizing a matrix, say $\mathbf{Z}$, the matrix may be accompanied by additional row or column data—"side information"—that may be able to "guide" the matrix factorization algorithm toward a more accurate or interpretable factorization. A recent prominent example of this in genomics research is spatial transcriptomics data [8], which is expression profiled in many genes at many spatial locations ("pixels") [9]. For a variety of reasons, one typically seeks to factorize $\mathbf{Z}$, the matrix of gene expression profiles. But the 2-d coordinates of the pixels also provide important information about the biological context of the cells; for example, we might expect nearby pixels to belong to the same cell type or tissue region. Therefore, "spatially aware" matrix factorization methods have recently been proposed for spatial transcriptomics data [10–12].

39th Conference on Neural Information Processing Systems (NeurIPS 2025).

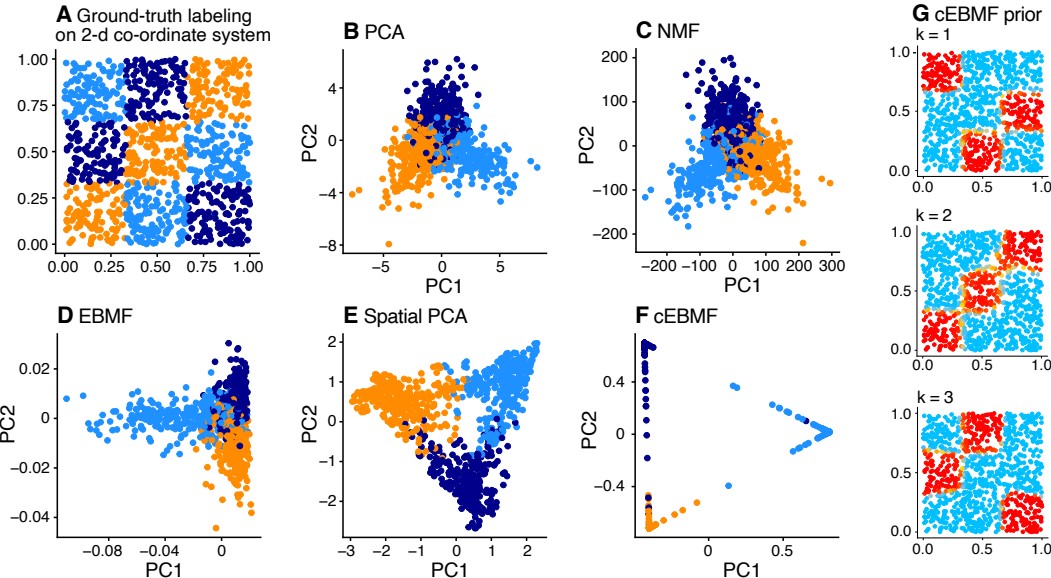

Figure 1: Toy simulation illustrating cEBMF for learning a matrix factorization, $\mathbf{Z} \approx \mathbf{LF}^T$. In this example, $\mathbf{Z}$ is a $1{,}000 \times 200$ matrix. Each of the $n = 1{,}000$ data points is assigned to one of three clusters (**orange**, **light blue**, **dark blue**). Points near each other tend to be assigned to the same cluster, except near the boundaries (A). Without the side information (the 2-d coordinates in A), PCA, NMF and EBMF with $K = 3$ factors cluster some points accurately, but many other points are not clustered accurately (B–D). By contrast, Spatial PCA [11] and our new method, cEBMF, which both incorporate the side information into the prior, more accurately cluster the points (E, F). (For consistency of visualization, the top 2 PCs of the $\mathbf{L}$ matrices from NMF, EBMF and cEBMF are shown.) Spatial PCA assumes the data points are spatial, whereas cEBMF does make this assumption; instead, it has a flexible prior that is adapted to the data. This learned prior is shown in G: the color of the points shows the prior probability that row $i$, column $k$ of $\mathbf{L}$ is nonzero (**blue** = low prior probability, **red** = high prior probability). See Sections 3 and 4 for definitions and additional details.

In this paper, we describe a novel matrix factorization framework that allows high-dimensional row and column data to guide the matrix factorizations without having to make specific assumptions about how these data inform the factorization. For example, although our framework can be applied to data that exhibit spatial properties, it does not assume or require that the data be spatial. Our framework is also flexible in that it includes many existing approaches as special cases, including unconstrained matrix factorization [13, 14], non-negative matrix factorization [15], semi-non-negative matrix factorization [16], and more recent methods that incorporate side information [17]. These features are achieved by taking an empirical Bayes approach, building on the recent empirical Bayes matrix factorization (EBMF) framework [13, 14]. In particular, we extend the EBMF approach of [13] with adaptive priors that are modified by the side information. We call this approach "covariate-moderated empirical Bayes matrix factorization," or "cEBMF" for short. See Fig. 1 for a toy example that illustrates the key features of cEBMF.

## 2 Related work

The literature on matrix factorization methods that incorporate side information is quite extensive. The different methods make different modeling assumptions, and are typically motivated by certain types of data. Although it is not possible to review all relevant literature here, we discuss a few of the most important or related methods.

Several variants of the topic model—which can be viewed as matrix factorizations with "sum-to-one" constraints on $\mathbf{L}$ and $\mathbf{F}$ [18]—incorporate side information in different ways; for example, the correlated topic model [19] and the structural topic model [20] incorporate document-level side information into the priors on $\mathbf{L}$. Collective matrix factorization (CMF) [21–23] has gained

considerable interest, but CMF is based on ideas that are quite different from cEBMF: like cEBMF, CMF assumes that the side information is the form of a matrix, but unlike cEBMF, CMF assumes that the side information factorizes in a similar way to $\mathbf{Z}$. Clearly, this assumption will not make sense for some types of data. Another prominent theme in matrix factorization with side information is incorporating group-level or categorical information, including ontological data. Among the methods in this area are CTPF [24] and the method of [25]. Another important class of methods related to cEBMF from the deep learning literature are variational autoencoders (VAE) [26], conditional variational autoencoders (cVAE) [27] and neural collaborative filtering (NCF) [28]. These methods generalize the concept of matrix factorization to nonlinear embeddings.

The method that is most closely related to cEBMF is MFAI [17] (see also [29] for related ideas). MFAI is in fact a special case of cEBMF in which the priors on $\mathbf{F}$ are normal and the prior means are informed by the covariates. Like cEBMF, MFAI allows these priors to be adapted separately for each dimension $k$. However, MFAI is not nearly as general as cEBMF; it implements only a single model, a single prior family with a specific parametric form, a specific procedure for fitting these priors (using gradient boosted tree methods [30]), and it only accommodates row-wise side information.

Several matrix factorization methods have been developed specifically for spatial transcriptomics data. Spatial PCA [11] models the spatial similarity among rows of $\mathbf{L}$ using Gaussian process prior. (Spatial PCA is similar to GP-LVM [31]. See also [32].) An NMF version of this approach generates "parts-based representations" guided by the spatial context of the data points [12]. More recently, IRIS [33] regularizes the matrix factors through a penalty function that encodes the spatial information in a graph (see also [34]).

## 3 Covariate-moderated empirical Bayes matrix factorization

### 3.1 Background: empirical Bayes matrix factorization

Empirical Bayes matrix factorization (EBMF) [13] is a flexible framework for matrix factorization: it approximates a matrix $\mathbf{Z} \in \mathbb{R}^{n \times p}$ as the product of two low-rank matrices,

$$\mathbf{Z} \approx \mathbf{L}\mathbf{F}^T, \tag{1}$$

where $\mathbf{L} \in \mathbb{R}^{n \times K}$, $\mathbf{F} \in \mathbb{R}^{p \times K}$, and $K \geq 1$. (In our applications, $K \ll n, p$.) EBMF assumes a normal model of the data,

$$\mathbf{Z} = \mathbf{L}\mathbf{F}^T + \mathbf{E}, \quad e_{ij} \sim N(0, \tau_{ij}^{-1}), \tag{2}$$

in which $N(\mu, \sigma^2)$ denotes the normal distribution with mean $\mu$ and variance $\sigma^2$, and the residual variances $\tau_{ij}^{-1}$ may vary by row ($i$) or by column ($j$) or both. (EBMF, and by extension cEBMF, also allows $\mathbf{Z}$ to contain missing values [13], which is important in many applications of matrix factorization, including collaborative filtering; see Sec. 4.2.) EBMF assumes prior distributions for elements of $\mathbf{L}$ and $\mathbf{F}$, which are themselves estimated among pre-specified prior families $\mathcal{G}_{\ell,k}$ and $\mathcal{G}_{f,k}$, respectively:

$$
\begin{aligned}
\ell_{ik} &\sim g_k^{(\ell)}, \quad g_k^{(\ell)} \in \mathcal{G}_{\ell,k}, \quad k = 1, \dots, K \\
f_{jk} &\sim g_k^{(f)}, \quad g_k^{(f)} \in \mathcal{G}_{f,k}, \quad k = 1, \dots, K.
\end{aligned}
\tag{3}
$$

The flexibility of EBMF comes from the wide range of possible prior families (including non-parametric families) [35]. Different choices of prior family correspond to different existing matrix factorization methods. For example, if all families $\mathcal{G}_{\ell,k}$ and $\mathcal{G}_{f,k}$ are the family of zero-mean normal priors, then $\mathbf{L}\mathbf{F}^T$ is similar to a truncated singular value decomposition (SVD) [36, 37]. When the prior families are all point-normal (mixture of a point mass at zero and a zero-centered normal), one obtains empirical Bayes versions of sparse SVD or sparse factor analysis [38–40]. The prior families can also constrain $\mathbf{L}$ and $\mathbf{F}$; for example, families that only contain distributions with non-negative support result in empirical Bayes versions of NMF. In summary, EBMF (2–3) is a highly flexible modeling framework for matrix factorization that includes important previous methods as special cases, but also many new combinations (e.g., [41]).

### 3.2 The cEBMF modeling framework

In covariate-moderated EBMF (cEBMF), we assume that we have some "side information" (covariates) for rows and/or columns of $\mathbf{Z}$ [42, 43]. Let $\boldsymbol{x}_i$ denote the available information for the $i$th row

of $\mathbf{Z}$, and let $\boldsymbol{y}_j$ denote the available information for the $j$th column of $\mathbf{Z}$. In principle, $\boldsymbol{x}_i$ and $\boldsymbol{y}_j$ can be any information processable by a neural net (text, graph, image, other structured data), but for simplicity we assume that this information is stored as a matrix. Therefore, let $\mathbf{X} \in \mathbb{R}^{n \times n_x}$ be a matrix containing information on the rows of $\mathbf{Z}$, with $\boldsymbol{x}_i$ corresponding to the $i$th row of $\mathbf{X}$ (e.g., $\boldsymbol{x}_i$ might contain the 2-d coordinate of cell $i$). Similarly, let $\mathbf{Y} \in \mathbb{R}^{p \times n_y}$ contain information on the columns of $\mathbf{Z}$, with $\boldsymbol{y}_j$ corresponding to the $j$th row of $\mathbf{Y}$. In cEBMF, we incorporate this side information into the model through *parameterized priors*,

$$
\begin{aligned}
\ell_{ik} &\sim g_k^{(\ell)}(\boldsymbol{x}_i), \quad g_k^{(\ell)}(\boldsymbol{x}_i) \in \mathcal{G}_{\ell,k}, \quad k = 1, \ldots, K \\
f_{jk} &\sim g_k^{(f)}(\boldsymbol{y}_j), \quad g_k^{(f)}(\boldsymbol{y}_i) \in \mathcal{G}_{f,k}, \quad k = 1, \ldots, K,
\end{aligned}
\tag{4}
$$

where $g_k^{(\ell)}(\boldsymbol{x}_i)$ is a probability distribution within the family $\mathcal{G}_{\ell,k}$, parameterized by $\boldsymbol{x}_i$, and $g_k^{(f)}(\boldsymbol{y}_j)$ is a probability distribution within the family $\mathcal{G}_{f,k}$ parameterized by $\boldsymbol{y}_i$.

A key limitation of many existing approaches is that they integrate the side information using restrictive parametric models that may or may not be appropriate for the particular application. Another limitation is that the priors chosen for these methods may make strong or perhaps unrealistic assumptions about the structure underlying the data; for example, Gaussian process priors, which have been used in matrix factorization (e.g., [11, 31, 44]), typically assume that the factors vary smoothly in space, which makes it difficult to capture sharp changes at boundaries [45]. Existing methods also typically rely on hyperparameters that need to be tuned or selected (e.g., using cross-validation).

To address these issues, we propose cEBMF, a method that:

1. Can leverage a large variety of models (e.g., multinomial regression, multilayer perceptron, graphical neural nets, convolutional neural nets) to integrate the side information into the prior.

2. Can use families of priors that are flexible in form and thus do not make strong assumptions.

3. Allows automatic selection of the hyperparameters in (4) via an empirical Bayes approach.

More formally, we fit a prior for each column $k$ of $\mathbf{L}$, which maps each vector of covariates $\boldsymbol{x}_i$ to a given element $g_k^{(\ell)}(\boldsymbol{x}_i) \in \mathcal{G}_{\ell,k}$, and similarly for each column $k$ of $\mathbf{F}$. In Sec. 3.3, we describe a simple yet general algorithm that simultaneously learns the factors $\mathbf{L}, \mathbf{F}$ and the priors $g_k^{(\ell)}(\boldsymbol{x}_i), g_k^{(f)}(\boldsymbol{y}_j)$. A PyTorch-based [46] implementation of cEBMF with several different parameterized prior families is available at `https://github.com/william-denault/cebmf_torch`.

### 3.2.1 An illustration: cEBMF with side information on factor sparsity

Here we illustrate the implementation of the cEBMF framework using a simple yet broadly applicable prior family. This prior family assumes that the covariates $\mathbf{X}, \mathbf{Y}$ *only inform the pattern of sparsity*—that is, the placement of zeros—in $\mathbf{L}$ and $\mathbf{F}$. This type of prior is of particular interest for matrix factorization because matrix factorizations are typically invariant to rescaling, and therefore priors that inform the magnitudes of $\ell_{ik}$ and $f_{jk}$ are difficult to design. (By "invariant to rescaling," we mean that the likelihood or objective does not change if we replace $\mathbf{L}\mathbf{F}^T$ by $\tilde{\mathbf{L}}\tilde{\mathbf{F}}^T$, where $\tilde{\mathbf{L}} = \mathbf{L}\mathbf{D}, \tilde{\mathbf{F}} = \mathbf{F}\mathbf{D}^{-1}$, and $\mathbf{D}$ is an invertible diagonal matrix.) We define this prior family as

$$
\mathcal{G}_{ss} := \{g : g(u) = (1 - \pi(\boldsymbol{x}, \boldsymbol{\theta}))\delta_0(u) + \pi(\boldsymbol{x}, \boldsymbol{\theta})g_1(u; \boldsymbol{\omega})\},
\tag{5}
$$

in which $\delta_0(u)$ denotes the point-mass at zero, $g_1(u; \boldsymbol{\omega})$ denotes the density of some probability distribution $g_1(\boldsymbol{\omega})$ on $u \in \mathbb{R}$, and $\boldsymbol{x} \in \mathbb{R}^m$ denotes the covariate. For example, when $g_1$ is the normal distribution and $\boldsymbol{\omega}$ specifies the mean and variance, (5) is a family of parameterized "spike-and-slab" priors [47], and cEBMF with $\mathcal{G}_{\ell,k} = \mathcal{G}_{ss}, \mathcal{G}_{f,k} = \mathcal{G}_{ss}$ implements a version of sparse factor analysis [38–40] in which the sparsity of the factors is informed by the covariates. (Note that the "ss" in $\mathcal{G}_{ss}$ is short for "spike-and-slab.") Alternatively, if $g_1$ is a distribution with support only on non-negative numbers, such as an exponential distribution, then cEBMF implements a version of sparse NMF. The free parameters are $\boldsymbol{\theta}$, which control the weight on the "spike", $\delta_0$, and $\boldsymbol{\omega}$, which control the shape of the "slab", $g_1$. One simple parameterization of $\pi(\boldsymbol{x}, \boldsymbol{\theta})$ uses a logistic regression model,

$$
\pi(\boldsymbol{x}, \boldsymbol{\theta}) = \phi\big(\theta_0 + \sum_{t=1}^m x_t \theta_t\big),
\tag{6}
$$

where $\phi(x) := 1/(1 + e^{-x})$ denotes the sigmoid function, and $\boldsymbol{\theta} \in \mathbb{R}^{m+1}$. Most of the parameterized prior families used in this paper and in the cEBMF software are variants or elaborations on $\mathcal{G}_{ss}$.

### 3.3 The cEBMF learning algorithm

A key feature of the cEBMF modeling framework is that the algorithm for fitting the priors and estimating the factorization is simple to describe and often straightforward to implement. In brief, the cEBMF learning algorithm reduces a complex model-fitting task to a series of simpler subproblems. Each of these subproblems involves fitting a covariate-moderated variant of an empirical Bayes normal means (EBNM) model [35]. This also has the advantage of making the cEBMF framework and software modular so that a method that solves a covariate-moderated EBNM problem can be "plugged in" to the generic cEBMF algorithm.

#### 3.3.1 Background: empirical Bayes normal means

Given observations $\hat{\beta}_i \in \mathbb{R}$ with known standard deviations $s_i > 0$, $i = 1, \ldots, n$, the normal means model [48–50] is

$$\hat{\beta}_i \overset{\text{ind.}}{\sim} N(\beta_i, s_i^2), \tag{7}$$

in which the "true" means $\beta_i \in \mathbb{R}$ are unknown. It is further assumed that the unknown means are

$$\beta_i \overset{\text{i.i.d.}}{\sim} g \in \mathcal{G}, \tag{8}$$

where $\mathcal{G}$ is some pre-specified family of probability distributions.

The empirical Bayes (EB) approach to fitting the normal means model (7–8) exploits the fact that the noisy observations $\hat{\beta}_i$ contain information not only about the underlying means $\beta_i$ but also how the means are collectively distributed. EB "borrows information" across the observations to estimate $g$; typically this is done by maximizing the marginal log-likelihood of (7–8). The unknown means $\beta_i$ are typically estimated by their posterior means (given the estimate of $g$).

To adapt the EBNM model (7–8) to cEBMF, we allow the prior for the $i$th unknown mean to depend on additional data $\boldsymbol{d}_i$ and parameters $\boldsymbol{\theta}$,

$$\beta_i \overset{\text{ind.}}{\sim} g(\boldsymbol{d}_i, \boldsymbol{\theta}) \in \mathcal{G}, \tag{9}$$

so that each combination of $\boldsymbol{\theta}$ and $\boldsymbol{d}_i$ maps to an element of $\mathcal{G}$. We refer to this as "covariate-moderated EBNM" (cEBNM). Solving the cEBNM problem therefore involves two key computations:

**1. Estimate the model parameters.** Compute

$$\hat{\boldsymbol{\theta}} := \underset{\boldsymbol{\theta} \in \mathbf{R}^m}{\operatorname{argmax}} \, \mathcal{L}(\boldsymbol{\theta}), \tag{10}$$

where $\mathcal{L}(\boldsymbol{\theta})$ denotes the marginal likelihood,

$$\mathcal{L}(\boldsymbol{\theta}) := p(\hat{\boldsymbol{\beta}} \mid \boldsymbol{s}, \boldsymbol{\theta}, \mathbf{D}) = \prod_{i=1}^{n} \int N(\hat{\beta}_i; \beta_i, s_i^2) \, g(\beta_i; \boldsymbol{d}_i, \boldsymbol{\theta}) \, d\beta_i, \tag{11}$$

in which $\hat{\boldsymbol{\beta}} = (\hat{\beta}_1, \ldots, \hat{\beta}_n)$, $\boldsymbol{s} = (s_1, \ldots, s_n)$, $\mathbf{D}$ is a matrix storing $\boldsymbol{d}_1, \ldots, \boldsymbol{d}_n$, $N(\hat{\beta}_i; \beta_i, s_i^2)$ denotes the density of $N(\beta_i, s_i^2)$ at $\hat{\beta}_i$, and $g(\beta_i; \boldsymbol{d}_i, \boldsymbol{\theta})$ denotes the density of $g(\boldsymbol{d}_i, \boldsymbol{\theta})$ at $\beta_i$.

**2. Compute posterior summaries.** Compute summaries from the posterior distributions given the estimated prior,

$$p(\beta_i \mid \hat{\beta}_i, s_i, \hat{\boldsymbol{\theta}}, \mathbf{D}) = \frac{N(\hat{\beta}_i; \beta_i, s_i^2) \, g(\beta_i; \boldsymbol{d}_i, \hat{\boldsymbol{\theta}})}{\int N(\hat{\beta}_i; t, s_i^2) \, g(t; \boldsymbol{d}_i, \hat{\boldsymbol{\theta}}) \, dt}. \tag{12}$$

For many classical prior families, such as the spike-and-slab family in Sec. 3.2.1, the integrals in (11) and (12) can be computed analytically. More generally, standard numerical techniques such as Gauss-Hermite quadrature may provide reasonably fast and accurate solutions for prior families that do not result in closed-form integrals since the integrals in (11) and (12) are one-dimensional. As a result, $\hat{\boldsymbol{\theta}}$ can often be obtained efficiently using off-the-shelf optimization algorithms even when the chosen priors do not result in analytical integrals.

In summary, solving the cEBNM problem consists of finding a mapping from the known quantities $(\hat{\boldsymbol{\beta}}, \boldsymbol{s}, \mathbf{D})$ to a tuple $(\hat{\boldsymbol{\theta}}, \hat{q})$, where each $(\boldsymbol{d}_i, \hat{\boldsymbol{\theta}})$ maps to an element $g(\boldsymbol{d}_i, \hat{\boldsymbol{\theta}}) \in \mathcal{G}$, and $\hat{q}$ is the posterior

distribution of the unknown means, $\hat{q}(\boldsymbol{\beta}) := \prod_{i=1}^{n} p(\beta_i \mid \hat{\beta}_i, s_i, \hat{\boldsymbol{\theta}}, \mathbf{D})$. To facilitate the description of the cEBMF algorithm below, we denote this mapping as

$$\text{cEBNM}(\hat{\boldsymbol{\beta}}, \boldsymbol{s}, \mathbf{D}, \mathcal{G}) = (\hat{\boldsymbol{\theta}}, \hat{q}). \qquad (13)$$

Note that in practice the full posterior $\hat{q}(\boldsymbol{\beta})$ is not needed; the first and second posterior moments are sufficient (see Sec. 3.3.2). Any prior family is admissible under the cEBMF framework so long as the mapping (13) is computable (either numerically or analytically).

### 3.3.2 Algorithm

Given a method for solving the cEBNM problem (Sec. 3.3.1), the cEBMF model can be fitted using a simple coordinate ascent algorithm. In brief, the cEBMF algorithm maximizes an objective function—the evidence lower bound (ELBO) [51] under a variational approximation with conditional independence assumptions on $\mathbf{L}$ and $\mathbf{F}$ (see the Appendix)—by iterating over the following updates for each factor $k = 1, \ldots, K$ until some stopping criterion is met:

1. Disregard the $k$th factor in $\bar{\mathbf{R}}$, the $n \times p$ matrix expected residuals, $\bar{\mathbf{R}}^k = \bar{\mathbf{R}} + \bar{\boldsymbol{\ell}}_k \bar{\boldsymbol{f}}_k^T$.

2. For each $i = 1, \ldots, n$, compute the least-squares estimates of $\ell_{ik}$, denoted $\hat{\ell}_{ik}$, and the standard deviations $s_{ik}^{\ell}$ of these estimates,

$$\hat{\ell}_{ik} = (s_{ik}^{\ell})^2 \sum_{j=1}^{p} \tau_{ij} \bar{r}_{ij}^k \bar{f}_{jk} \qquad (14)$$

$$s_{ik}^{\ell} = \left( \sum_{j=1}^{p} \tau_{ij} \bar{f}_{jk}^2 \right)^{-1/2}, \qquad (15)$$

where $\bar{f}_{jk}$ and $\bar{f}_{jk}^2$ denote, respectively, the first and second posterior moments of $f_{jk}$.

3. Update $g_k^{(\ell)} \in \mathcal{G}_{\ell,k}$ by solving (10), in which we make the following substitutions in (10): $\hat{\beta}_i \leftarrow \hat{\ell}_{ik}, s_i \leftarrow s_{ik}^{\ell}, i = 1, \ldots, n, \mathbf{D} \leftarrow \mathbf{X}, \mathcal{G} \leftarrow \mathcal{G}_{\ell,k}$.

4. Making the same substitutions in (12), update the posterior means $\bar{\boldsymbol{\ell}}_k = (\bar{\ell}_{1k}, \ldots, \bar{\ell}_{nk})^T$ and posterior second moments $\bar{\boldsymbol{\ell}}_k^2 = (\bar{\ell}_{1k}^2, \ldots, \bar{\ell}_{nk}^2)^T$.

5. Perform updates similar to those in Steps 2–4 to update $\bar{\boldsymbol{f}}_k$, $\bar{\boldsymbol{f}}_k^2$ and $g_k^{(f)} \in \mathcal{G}_{f,k}$.

6. Update the matrix of expected residuals, $\bar{\mathbf{R}} = \bar{\mathbf{R}}^k - \bar{\boldsymbol{\ell}}_k \bar{\boldsymbol{f}}_k^T$.

These steps are iterated until some stopping criterion is met. The algorithm must be initialized with initial estimates of $\bar{\mathbf{L}}, \bar{\mathbf{F}}$. The expected residuals are then initialized as $\bar{\mathbf{R}} = \mathbf{Z} - \bar{\mathbf{L}}\bar{\mathbf{F}}^T$. Note that to simplify presentation we have omitted some details such as how to update the residual variances $\tau_{ij}^{-1}$. These and other details are provided in the Appendix.

## 4 Experiments

### 4.1 Simulations

To assess the benefits of cEBMF, we compared cEBMF with other matrix factorization methods in simulated data sets. We compared with several methods that do not use side information, including EBMF (flashier R package [13, 35]), penalized matrix decomposition ("PMD"; PMA R package [39]), and a variational autoencoder (VAE) [26] implemented in PyTorch [46]. We also compared with other methods that use side information, including MFAI (mfair R package [17]), Spatial PCA [11], conditional VAE (cVAE) [27], and neural collaborative filtering (NCF) [28]. cVAE and NCF were also implemented in PyTorch. Note that Spatial PCA accepts only a specific type of side information, the 2-d coordinates of the data points, so was not included in all the simulations.

We compared the methods in four simulation scenarios designed to capture a range of settings where one might perform a matrix factorization analysis, with or without side information: (1) a "sparsity-driven covariate" setting in which the covariates only informed the sparsity pattern of $\mathbf{L}$ and $\mathbf{F}$; (2) an "uninformative covariate" setting in which the covariates provided no information about the true matrix factorization; (3) a "tiled-clustering" setting in which $\mathbf{L}$ depended on the 2-d location of the data points; and (4) a "shifted tiled-clustering" setting in which the cEBMF priors were unable recover the true data generating process. The latter scenario was used to assess cEBMF under model

misspecification. We simulated 100 data sets in each setting, and we assessed the ability of each method to recover the true matrix factorization as measured by root mean squared error (RMSE) between the true matrix factorization $\mathbf{LF}^T$ and estimated matrix factorization $\hat{\mathbf{L}}\hat{\mathbf{F}}^T$. More detailed descriptions of the simulations and the methods compared are given in the Appendix.

The results are summarized in Fig. 2. cEBMF was generally more accurate than the other methods, particularly when the covariates were informative; cEBMF achieved the greatest gains over EBMF in the tiled-clustering setting where the covariates were also the most informative. Reassuringly, cEBMF did not perform worse than EBMF in settings with an uninformative covariate or a prior that was misspecified ("shifted tiled-clustering"). The deep learning approaches (VAE, cVAE, NCF) generally performed worse than the other methods. cVAE sometimes outperformed VAE when the side information was highly informative, such as in the tiled-clustering scenario, but did not provide improvements over VAE in the more challenging sparsity-driven scenario. We also ran Spatial PCA on the tiled-clustering and shifted tiled-clustering data sets where the factors were partly driven by the 2-d locations of the data points. Despite the fact that Spatial PCA can exploit the side information, it had worse accuracy than EBMF which did not

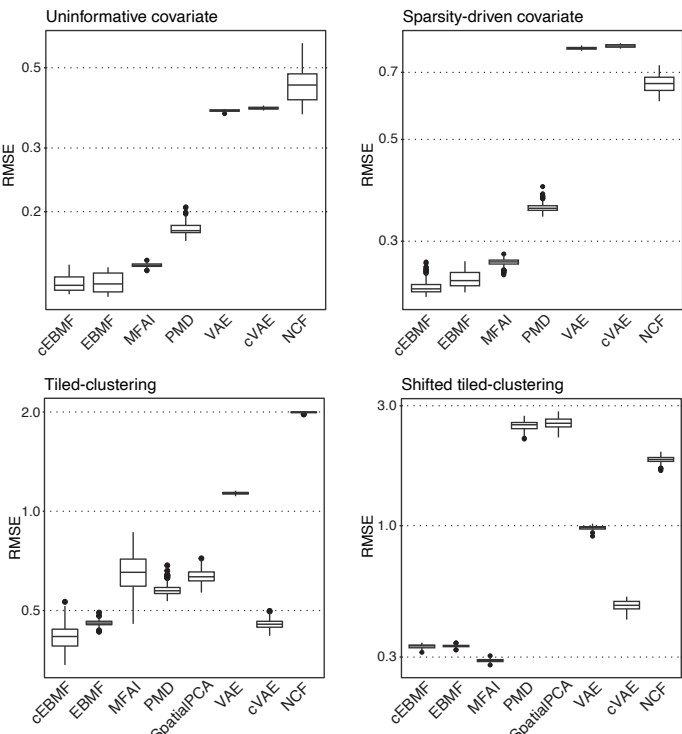

Figure 2: Performance of the different matrix factorization methods in the simulated data sets. Each boxplot summarizes the root mean squared errors (RMSEs) across 100 simulations in that scenario. (Lower RMSEs are better.) See Figures 5–8 in the Appendix for additional results from the simulations. Note Fig. 1 shows results from one of the tiled-clustering simulations in detail.

use the side information. This may be because Spatial PCA makes assumptions (e.g., orthogonal factors) that were not met by our simulations. MFAI generally performed worse than EBMF and cEBMF except in the shifted tiled-clustering setting; MFAI is a much less flexible model than cEBMF and therefore its performance was sensitive to the appropriateness of its modeling assumptions. (All models were misspecified in the shifted tiled-clustering setting, but perhaps MFAI was the least misspecified.) Additional results including comparisons with other methods (PCA/SVD, Sparse SVD [38], CMF [21]) are in the Appendix.

## 4.2 Collaborative filtering

To provide a quantitative assessment of the matrix factorization methods in real data, we ran the same methods on the MovieLens 100K data [52], a standard collaborative filtering benchmark in which the goal is to predict the unobserved elements of the matrix. Here, $\mathbf{Z}$ was a $1,682 \times 943$ matrix containing integer-valued movie ratings, with rows corresponding to movies and columns corresponding to users. Since most (93%) of the movie ratings were missing, this example highlights the ability of these methods to handle missing data (unlike most NMF methods). The side information $\mathbf{X}$ was a $1,682 \times 19$ binary matrix containing information about the movie's genre (comedy, adventure, etc). We held out some of the moving ratings at random, and used these held-out ratings as a test set.

We ran EBMF and cEBMF so as to produce non-negative matrix factorizations, which is common in collaborative filtering (e.g., [21]). Therefore, in the results we labeled these methods as "EBNMF" and "cEBNMF". (Note that MFAI cannot produce non-negative matrix factorizations.) To enforce non-

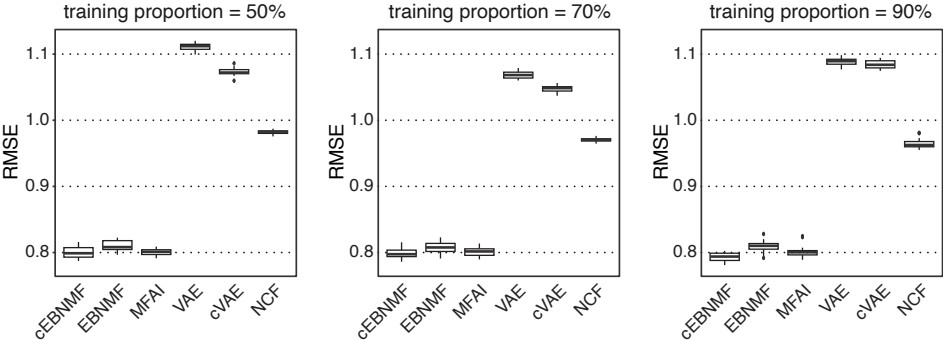

Figure 3: Prediction performance of different matrix factorization and deep learning methods in the MovieLens 100K data [52]. Training proportion = $X\%$ means that $X\%$ of the movie ratings were used in training, and the remaining $(100 - X)\%$ were used to evaluate accuracy (measured using RMSE). The results at each training proportion are from 10 random training-test splits.

negativity in $\mathbf{L}$ and $\mathbf{F}$, we used mixture-of-exponentials priors. For cEBNMF, the side information was incorporated into the priors on $\mathbf{L}$ using a multi-layer perceptron. (See the Appendix for further details.) Since we didn't know the true number of factors, $K$ was chosen adaptively in EBMF, cEBMF and MFAI. (We set an upper limit of 7 for EBMF and cEBMF, and 12 for MFAI.)

The results are summarized in Fig. 3. Both MFAI and cEBNMF were able to use the side information (the movie genres) to improve over EBNMF, and all three matrix factorization methods were more accurate than the deep learning methods. We conjecture that the deep learning methods would have performed better with more data (such as the more recently released MovieLens data sets that are much larger). On the MovieLens 100K data, cEBNMF yielded overall the best prediction accuracy across the different training-set splits.

## 4.3 Spatial transcriptomics

Although cEBMF was not specifically designed for spatial data, here we show that cEBMF also yields compelling results from spatial transcriptomics data [8] by exploiting the side information, the spatial locations of the data points. We illustrate this using a data set [53, 54] that has been annotated by domain experts and has been used in several papers to benchmark methods for spatial transcriptomics (e.g., [11, 55–57]). The data were collected from 12 slices of the human dorsolateral prefrontal cortex (DLPFC). After data preprocessing, each slice contained about 4,000 pixels and expression measured in about 5,000 genes ($n \approx 4000$, $p \approx 5{,}000$).[1]

In this application, our aim was to generate a "parts-based" representation of the data, with the hopes that the "parts" would resolve to biologically interpretable units (e.g., cell types, tissue regions, gene programs) [12, 58, 59]. This is a fundamentally different aim from the previous examples: in the previous examples, the goal was to generate accurate low-dimensional representations, but we did not ask whether the *individual dimensions* were accurate or interpretable. With this aim in mind, we ran cEBMF so as to produce non-negative matrix factorizations ("cEBNMF") using the same priors that were used for the MovieLens data. We compared to two other non-negative matrix factorizations that did not leverage the side information: NMF implemented in the R package NNLM [15], and EBMF with point-exponential priors ("EBNMF"). (The point-exponential prior is a simplification of the mixture-of-exponentials prior with a single exponential component in which the rate parameter is also learned.). We also compared to several of the methods that were considered in the previous experiments, including methods such as Spatial PCA and cVAE that make use of the side information, and others that do not.

Spatial PCA deserves special mention because it was specifically designed for spatial transcriptomics data [11]. Although Spatial PCA does not produce a parts-based decomposition, the Spatial PCA software automatically clusters the data points after projection onto the principal components (PCs), and this clustering can be compared to the non-negative matrix factorizations. Following [11], we

---

[1]We used the data prepared by the authors of the SpatialLIBD package [53] which were made available for download at `https://research.libd.org/spatialLIBD/`.

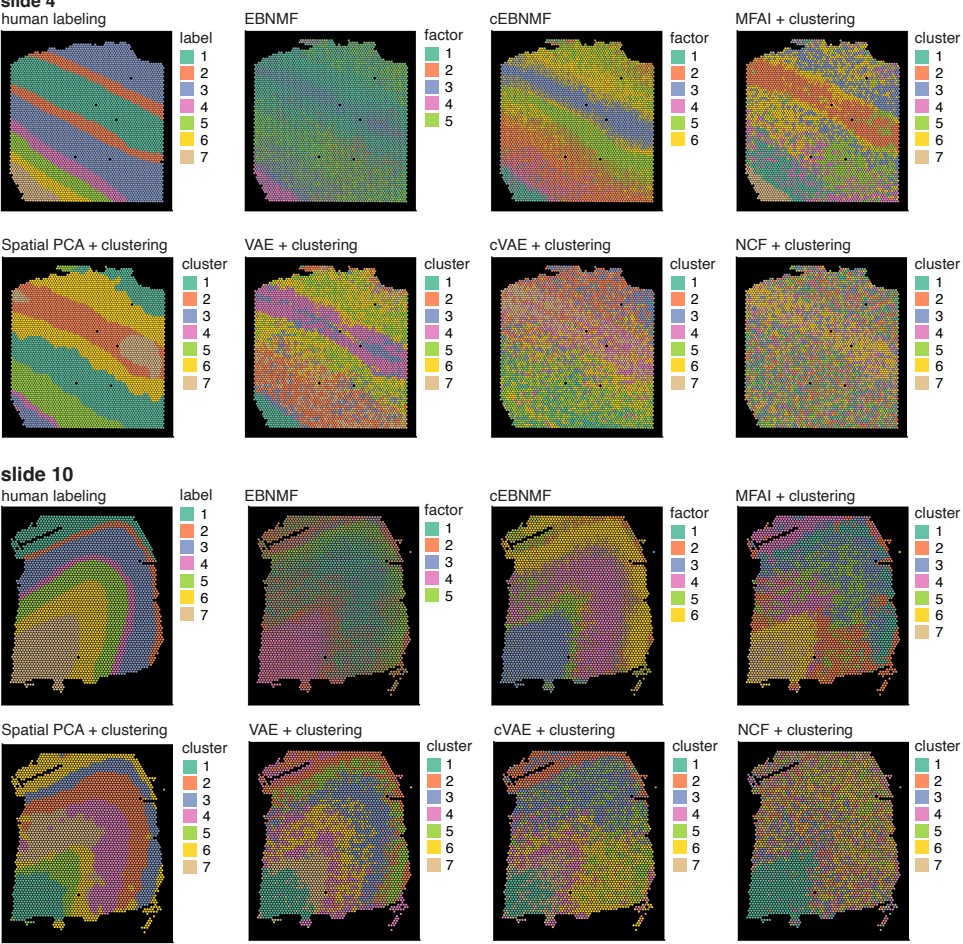

Figure 4: Results on slides 4 (top row) and 10 (bottom row) of the DLPFC spatial transcriptomics data [53, 54]. For the NMF, EBMF and cEBMF results, each data point ("pixel") $i$ is shown as a pie chart using the relative values of $i$th row of $\mathbf{L}$ (after performing an "LDA-style" post-processing of $\mathbf{L}, \mathbf{F}$ [12]). (Higher-resolution versions of these images are available online at `https://github.com/william-denault/cEBMF_experiment`.) Since MFAI and other methods did not produce a non-negative matrix factorization, we clustered the low-dimensional embeddings using the same approach that was used in Spatial PCA [11]. CMF results for these two slices and additional results on all 12 slices are given in the Appendix.

computed the top 20 PCs, then we ran the walk-trap clustering algorithm [60] on the PCs. Additional details of the Spatial PCA analysis and the other methods are given in the Appendix.

Figure 4 shows results on two of the slices, with additional results on all 12 slices provided in the Appendix. The manual annotations on the left-hand side should be viewed as a useful reference point, but not necessarily the "ground truth". (Consider that the data-driven annotations might identify previously unknown cellular structures.) EBNMF, cEBNMF and MFAI adapted the number of factors $K$ to the data (with upper limits of 50, 20 and 9, respectively). For the other methods, the number of clusters was set to match the manual annotation. Qualitatively, some of the factors from NMF and EBMF seem to correspond to the expert-labeled regions, but several other factors appear to be capturing other substructures that have no obvious spatial quality. Comparatively, the cEBNMF results in slices 4 and 9 capture the expert labeling much more closely, with most factors showing a clear spatial quality. The clusters from Spatial PCA, MFAI and VAE also capture spatial structure and expert labeling well, but with some notable exceptions, e.g., Spatial PCA cluster 7 in slices 4 and 9. (The Spatial PCA software performed additional post-processing on the clusters which is why these clusters look less "noisy" than the others.) cVAE, despite using the side information, did not seem to

| method | software | number of rows ($n$) | | | |
|--------|----------|------|------|------|------|
| | | $10^3$ | $10^4$ | $10^5$ | $10^6$ |
| EBMF | flashier [13, 35] | 0.8 | 2.5 | 36.9 | 165.1 |
| cEBMF | cebmf_torch* | 5.2 | 35.5 | 416.3 | 3,403.6 |
| MFAI | mfair [17] | 45.4 | 251.3 | 11,293.2 | – |
| Spatial PCA | SpatialPCA [11] | 234.8 | 8,213.7 | – | – |

Table 1: Running times of matrix factorization methods on data sets in which we varied $n$, the number of rows in $\mathbf{X}$ and $\mathbf{Z}$. The numbers in the table are the average running times (in seconds) from 10 simulated data sets. *Available at `https://github.com/william-denault/cebmf_torch`.

improve over VAE. The CMF results were comparatively poor (Fig. 9 in the Appendix), reflecting the inappropriateness of the CMF assumptions in this setting. Note that the NMF methods can capture continuous variation in expression within and across cell types or regions—as well as the expectation that some pixels might reflect combinations of cellular structures—whereas the clustering cannot.

### 4.4 Scalability benchmark

cEBMF can also handle much larger data sets than the MovieLens and DLPFC data sets considered above. (One reason we did not use larger data sets was to allow for comparison with methods that do not scale well to large data sets.) To illustrate this, we ran EBMF and cEBMF on "tiled-clustering" data sets (using the same priors described Sec. 4.1), in which the data sets were simulated with different numbers of rows, $n$. We compared the running times with two other matrix factorization methods that make use of the side information, MFAI and Spatial PCA (Table 1). While cEBMF had considerably higher running times than EBMF on the same data, cEBMF completed in much less time on average than MFAI and Spatial PCA. Further, while cEBMF was able to handle data sets with 1 million rows, Spatial PCA struggled to analyze data sets with 100,000 or more rows due to its high memory usage; for example, Spatial PCA needed approximately 300 GB of memory for $n = 100,000$. MFAI crashed frequently in data sets with $n \geq 100,000$ rows (only 2 of 10 of the runs completed at $n = 100,000$). Note this benchmark was performed on a computer with 32 GB memory, an NVIDIA GeForce RTX[TM] 4070 GPU and an AMD Ryzen[TM] 9 7940HS CPU (8 cores, 16 threads). The EBMF and cEBMF algorithms were run for at most 20 iterations. See also the Appendix where we describe some of the computational properties of the cEBMF algorithmic framework.

## 5 Conclusions

We have introduced cEBMF, a general and flexible framework for matrix factorization that (i) incorporates side information through flexible covariate-dependent priors and (ii) learns these priors from the data using empirical Bayes ideas. Considerable effort has gone into optimizing the software implementation building on our previous work on this topic [13, 35]. As a result, cEBMF scales well to large data sets with, say, hundreds of thousands or millions of rows and/or columns. Our experiments highlight the importance of using matrix factorization models that make appropriate assumptions about the data or are sufficiently flexible to adapt to the data. In our experiments, cEBMF performed competitively against other matrix factorization methods and deep learning approaches that make use of the side information. Because the priors in cEBMF can take the form of virtually any probabilistic model optimized via equations (10–11), our framework opens the door to incorporating other types of side information, including images and graphs.

**Note:** R and Python code implementing the experiments is available at `https://github.com/william-denault/cEBMF_experiment`, and a PyTorch-based implementation of cEBMF is available at at `https://github.com/william-denault/cebmf_torch`.

## Acknowledgments

We thank the staff at the Research Computing Center at the University of Chicago for providing the high-performance computing resources used to implement the numerical experiments. We also thank

Deaglan Bartlett and Augustin Marignier for their help in developing the PyTorch implementation. This work was supported in part by NIH grant R01HG002585 (to M.S.) and Eric and Wendy Schmidt AI in Science Postdoctoral Fellowship, a Schmidt Sciences, LLC program (to W.R.P.D.). Additional support came from the University of Chicago Data Science Institute through the 2024 AI+Science Research Initiative. We also thank the reviewers for their feedback.

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

# Appendices

## A   Derivations and additional method details

The cEBMF algorithm (Sec. 3.3.2) is a *variational empirical Bayes* algorithm [61–64] that is formulated as solving the following optimization problem:

$$\underset{q,\,g,\,\boldsymbol{\tau}}{\operatorname{argmax}}\,\text{ELBO}(q,g,\boldsymbol{\tau}), \tag{16}$$

where $g$ is shorthand for the priors $g_1^{(\ell)},\dots,g_K^{(\ell)},g_1^{(f)},\dots,g_K^{(f)}$, $q$ is a distribution on $(\mathbf{L},\mathbf{F})$, and $\text{ELBO}(q,g,\boldsymbol{\tau})$ is the "Evidence Lower BOund" (ELBO) [51], a lower bound to the "evidence", $\log p(\mathbf{Z}\mid g,\boldsymbol{\tau})$:

$$\text{ELBO}(q,g,\boldsymbol{\tau}) := \mathbb{E}_q[\log p(\mathbf{Z}\mid \mathbf{L},\mathbf{F},\boldsymbol{\tau})] + \mathbb{E}_q\left[\log\left\{\frac{p(\mathbf{L},\mathbf{F}\mid\mathbf{X},\mathbf{Y})}{q(\mathbf{L},\mathbf{F})}\right\}\right]. \tag{17}$$

See [13, 65, 66] for other variational empirical Bayes algorithms derived in a similar way.

To achieve tractable update expressions for the model parameters, we approximate the posterior $q(\mathbf{L},\mathbf{F})$ so that it factorizes over all elements of $\mathbf{L}$ and all elements of $\mathbf{F}$ (sometimes called a "mean field" approximation):

$$
\begin{aligned}
q(\mathbf{L},\mathbf{F}) &= q^\ell(\mathbf{L})\,q^f(\mathbf{F}) \\
q^\ell(\mathbf{L}) &= \prod_{i=1}^{n}\prod_{k=1}^{K} q_{ik}^\ell(\ell_{ik}) \\
q^f(\mathbf{F}) &= \prod_{j=1}^{p}\prod_{k=1}^{K} q_{jk}^f(f_{jk}).
\end{aligned}
\tag{18}
$$

With this factorization (or conditional independence) qconstraint on $q$, the right-hand part of the ELBO can be immediately decomposed into a sum of expectations over the individual elements of $\mathbf{L}$ and $\mathbf{F}$, so we have

$$
\begin{aligned}
\text{ELBO}(q,g,\boldsymbol{\tau}) = &\,\mathbb{E}_q[\log(p(\mathbf{Z}\mid\mathbf{L},\mathbf{F},\boldsymbol{\tau})] \\
&+ \sum_{i=1}^{n}\sum_{k=1}^{K}\mathbb{E}_q\left[\log\left\{\frac{g_k^{(\ell)}(l_{ik};\boldsymbol{x}_i)}{q_{ik}^\ell(\ell_{ik})}\right\}\right] \\
&+ \sum_{j=1}^{p}\sum_{k=1}^{K}\mathbb{E}_q\left[\log\left\{\frac{g_k^{(f)}(f_{jk};\boldsymbol{y}_j)}{q_{jk}^f(f_{jk})}\right\}\right],
\end{aligned}
\tag{19}
$$

where $g_k^{(\ell)}(\ell;\boldsymbol{x}_i)$ denotes the density of $g_k^{(\ell)}(\boldsymbol{x}_i)$ at $\ell$, and $g_k^{(f)}(f;\boldsymbol{y}_j)$ denotes the density of $g_k^{(f)}(\boldsymbol{x}_i)$ at $f$.

### A.1   Updating the factors

The following proposition formally connects the updates of the individual factors $k=1,\dots,K$ (Step 2–4 of the algorithm in Sec. 3.3.2) to learning a covariate-moderated EBNM model (Sec. 3.3.1).

**Proposition 1.** Let $\boldsymbol{\ell}_k = (\ell_{1k},\dots,\ell_{nk})^T$ denote the $k$th column of $\mathbf{L}$, let $\boldsymbol{f}_k = (f_{1k},\dots,f_{pk})^T$ denote the $k$th column of $\mathbf{F}$, let $\bar{\boldsymbol{\ell}}_k = \mathbb{E}_q[\boldsymbol{\ell}_k]$, $\bar{\boldsymbol{f}}_k = \mathbb{E}_q[\boldsymbol{f}_k]$, $\bar{\boldsymbol{\ell}}_k^2 = \mathbb{E}_q[\boldsymbol{\ell}_k^2]$ and $\bar{\boldsymbol{f}}_k^2 = \mathbb{E}_q[\boldsymbol{f}_k^2]$, and we further define

$$q_k^\ell(\boldsymbol{\ell}_k) := \prod_{i=1}^{n} q_{ik}^\ell(\ell_{ik}) \tag{20}$$

$$q_k^f(\boldsymbol{f}_k) := \prod_{j=1}^{p} q_{jk}^f(f_{jk}). \tag{21}$$

Let $\bar{\mathbf{R}}^k$ denote the $n \times p$ matrix of expected residuals (with elements $\bar{r}_{ij}^k$) that ignores the contribution of the $k$th factor,

$$\bar{r}_{ij}^k := z_{ij} - \sum_{k' \neq k} \bar{l}_{ik'} \bar{f}_{jk'}. \tag{22}$$

Also define $\hat{\boldsymbol{\ell}}(\mathbf{Z}, \boldsymbol{t}, \boldsymbol{w}, \boldsymbol{\tau})$, $\hat{\boldsymbol{f}}(\mathbf{Z}, \boldsymbol{t}, \boldsymbol{w}, \boldsymbol{\tau})$, $\boldsymbol{s}^\ell(\boldsymbol{w}, \boldsymbol{\tau})$ and $\boldsymbol{s}^f(\boldsymbol{w}, \boldsymbol{\tau})$ as vector-valued functions in which the individual vector elements given by

$$\hat{\ell}_i(\mathbf{Z}, \boldsymbol{t}, \boldsymbol{w}, \boldsymbol{\tau}) = \frac{\sum_{j=1}^p \tau_{ij} z_{ij} t_j}{[s_i^\ell(\boldsymbol{w}, \boldsymbol{\tau})]^2} \tag{23}$$

$$\hat{f}_j(\mathbf{Z}, \boldsymbol{t}, \boldsymbol{w}, \boldsymbol{\tau}) = \frac{\sum_{i=1}^n \tau_{ij} z_{ij} t_i}{[s_j^f(\boldsymbol{w}, \boldsymbol{\tau})]^2} \tag{24}$$

$$s_i^\ell(\boldsymbol{w}, \boldsymbol{\tau}) = (\textstyle\sum_{j=1}^p \tau_{ij} w_j)^{-1/2} \tag{25}$$

$$s_j^f(\boldsymbol{w}, \boldsymbol{\tau}) = (\textstyle\sum_{i=1}^n \tau_{ij} w_i)^{-1/2}. \tag{26}$$

Then using the definition of the ELBO in (17) and the cEBNM mapping defined in (13), we have that

$$\operatorname{argmax}_{q_k^\ell, g_k^{(\ell)} \in \mathcal{G}_{\ell,k}} \operatorname{ELBO}(q, g, \boldsymbol{\tau}) = \operatorname{cEBNM}(\hat{\boldsymbol{\ell}}(\bar{\mathbf{R}}^k, \bar{\boldsymbol{f}}_k, \bar{\boldsymbol{f}}_k^2, \boldsymbol{\tau}), \boldsymbol{s}_\ell(\bar{\boldsymbol{f}}_k^2, \boldsymbol{\tau}), \mathbf{X}, \mathcal{G}_{\ell,k}) \tag{27}$$

$$\operatorname{argmax}_{q_k^f, g_k^{(f)} \in \mathcal{G}_{f,k}} \operatorname{ELBO}(q, g, \boldsymbol{\tau}) = \operatorname{cEBNM}(\hat{\boldsymbol{f}}(\bar{\mathbf{R}}^k, \bar{\boldsymbol{\ell}}_k, \bar{\boldsymbol{\ell}}_k^2, \boldsymbol{\tau}), \boldsymbol{s}_f(\bar{\boldsymbol{\ell}}_k^2, \boldsymbol{\tau}), \mathbf{Y}, \mathcal{G}_{f,k}). \tag{28}$$

Note that this identity requires a slight change to the definition of the cEBNM mapping (13) as returning the priors $g(\boldsymbol{d}_i, \hat{\boldsymbol{\theta}})$ at $\hat{\boldsymbol{\theta}}$ rather than the parameter estimates $\hat{\boldsymbol{\theta}}$ themselves.

*Proof.* Starting from (19), we expand on the parts of ELBO that involve $q_k^\ell$ or $g_k^{(\ell)}$ or both:

$$\operatorname{ELBO}(q, g, \boldsymbol{\tau}) = -\frac{1}{2} \sum_{i=1}^n \mathbb{E}_q[a_{ik} l_{ik}^2 - 2b_{ik} l_{ik}] + \sum_{i=1}^n \mathbb{E}_q\left[\log\left\{\frac{g_k^{(\ell)}(\ell_{ik}; \boldsymbol{x}_i)}{q_{ik}^{(\ell)}(\ell_{ik})}\right\}\right] + \text{const}, \tag{29}$$

where "const" is a placeholder for the terms in the ELBO that do not depend on the $k$th factor, and we define

$$a_{ik} := \sum_{j=1}^p \tau_{ij} (\bar{f}_{jk})^2 \tag{30}$$

$$b_{ik} := \sum_{j=1}^p \tau_{ij} \bar{r}_{ij}^k \bar{f}_{jk}. \tag{31}$$

The identity (27) then follows from Lemma 1 (given below). The other identity (28) is proved similarly. □

## A.2  Updating the residual variances

Focussing on the part of the ELBO depends on $\boldsymbol{\tau}$, we have

$$\operatorname{ELBO}(q, g, \boldsymbol{\tau}) = \frac{1}{2} \sum_{i=1}^n \sum_{j=1}^p (\log \tau_{ij} - \tau_{ij} \bar{r}_{ij}^2) + \text{const}, \tag{32}$$

in which "const" is a placeholder for the other terms in the ELBO that do not involve $\boldsymbol{\tau}$, and $\bar{r}_{ij}^2$ is the expected squared difference between the observation $z_{ij}$ and the value predicted by the matrix factorization:

$$\bar{r}_{ij}^2 := \mathbb{E}_q[(z_{ij} - \hat{z}_{ij})^2], \tag{33}$$

where

$$\hat{z}_{ij} = \sum_{k=1}^K l_{ik} f_{jk}. \tag{34}$$

If one makes the modeling assumption that all the residual variances are the same, i.e., $\tau_{ij} = \tau$, then from (32) the update for $\tau$ works out to

$$\tau = \frac{n \times p}{\sum_{i=1}^{n} \sum_{j=1}^{p} \bar{r}_{ij}^2}. \tag{35}$$

If instead one makes the weaker modeling assumption that the residual variances are the same in each column, i.e., $\tau_{ij} = \tau_j, j = 1, \ldots, p$, then the updates work out to

$$\tau_j = \frac{n}{\sum_{i=1}^{n} \bar{r}_{ij}^2}. \tag{36}$$

Similarly, for row-specific residual variances the updates are

$$\tau_i = \frac{p}{\sum_{j=1}^{p} \bar{r}_{ij}^2}. \tag{37}$$

For all these expressions, the squared differences $\bar{r}_{ij}^2$ are easily computed given the conditional independence assumptions of the fully-factorized approximation (18):

$$\bar{r}_{ij}^2 = \left( z_{ij} - \sum_{k=1}^{K} \bar{l}_{ik} \bar{f}_{jk} \right)^2 + \sum_{k=1}^{K} (\bar{l}_{ik}^2)(\bar{f}_{jk}^2) - \sum_{k=1}^{K} (\bar{l}_{ik} \bar{f}_{jk})^2, \tag{38}$$

in which we have defined $\bar{l}_{ik} := \mathbb{E}_q[l_{ik}]$, $\bar{f}_{jk} := \mathbb{E}_q[f_{jk}]$, $\bar{l}_{ik}^2 := \mathbb{E}_q[l_{ik}^2]$ and $\bar{f}_{jk}^2 := \mathbb{E}_q[f_{jk}^2]$.

## A.3 Covariate-moderated EBNM

To complete the proof of Proposition 1, it remains to show that the identity (27) is satisfied at the objective function given by (29). (And similarly for the identity (28).) This connection is made in the following lemma.

**Lemma 1.** Consider the cEBNM mapping defined in (13). An equivalent definition of this mapping is

$$(\hat{\boldsymbol{\theta}}, \hat{q}) = \operatorname{argmax}_{\boldsymbol{\theta}, q} F(\boldsymbol{\theta}, q; \hat{\boldsymbol{\beta}}, \boldsymbol{s}, \mathbf{D}), \tag{39}$$

where

$$F(\boldsymbol{\theta}, q; \hat{\boldsymbol{\beta}}, \boldsymbol{s}, \mathbf{D}) = -\frac{1}{2} \sum_{i=1}^{n} \mathbb{E}_q[a_i \beta_i^2 - 2b_i \beta_i] + \sum_{i=1}^{n} \mathbb{E}_q \left[ \log \left\{ \frac{g(\beta_i; \boldsymbol{d}_i, \boldsymbol{\theta})}{q_i(\beta_i)} \right\} \right], \tag{40}$$

and $g(\beta; \boldsymbol{d}_i, \boldsymbol{\theta})$ denotes the density of $g(\boldsymbol{d}_i, \boldsymbol{\theta})$ at $\beta$, and we further define

$$q(\boldsymbol{\beta}) = \prod_{i=1}^{n} q_i(\beta_i) \tag{41}$$

$$a_i = 1/s_i^2 \tag{42}$$

$$b_i = \hat{\beta}_i/s_i^2. \tag{43}$$

*Proof.* We begin with the ELBO for the cEBNM model (7, 9):

$$\text{ELBO}(\boldsymbol{\theta}, q; \hat{\boldsymbol{\beta}}, \boldsymbol{s}, \mathbf{D}) = \log \mathcal{L}(\boldsymbol{\theta}) - D_{\text{KL}}(q \,\|\, p_{\text{post}}). \tag{44}$$

where $D_{\text{KL}}(q \,\|\, p)$ denotes the Kullback-Leibler (K-L) divergence from a distribution $p$ to a distribution $q$ [67], $\mathcal{L}(\boldsymbol{\theta})$ is the marginal likelihood defined in (11), and $p_{\text{post}}(\boldsymbol{\beta})$ is the (exact) posterior distribution, $p_{\text{post}}(\boldsymbol{\beta}) := \prod_{i=1}^{n} p(\beta_i \mid \hat{\beta}_i, s_i, \hat{\boldsymbol{\theta}}, \mathbf{D})$ (see eq. 12). Since $D_{\text{KL}}(q \,\|\, p)$ is always zero or greater, and is exactly zero when $p = q$, we have that $\operatorname{argmax}_q \text{ELBO}(\boldsymbol{\theta}, q; \hat{\boldsymbol{\beta}}, \boldsymbol{s}, \mathbf{D}) = p_{\text{post}}$ and $\max_q \text{ELBO}(\boldsymbol{\theta}, q; \hat{\boldsymbol{\beta}}, \boldsymbol{s}, \mathbf{D}) = \log \mathcal{L}(\boldsymbol{\theta})$. Next, a basic identity of the ELBO (see for example Appendix B of [68]) is that the ELBO can be rewritten as

$$mathrm{ELBO}(\boldsymbol{\theta}, q; \hat{\boldsymbol{\beta}}, \boldsymbol{s}, \mathbf{D}) = \mathbb{E}_q[\log p(\hat{\boldsymbol{\beta}} \mid \boldsymbol{\beta}, \boldsymbol{s})] + \sum_{i=1}^{n} \mathbb{E}_q \left[ \log \left\{ \frac{g(\beta_i; \boldsymbol{d}_i, \boldsymbol{\theta})}{q_i(\beta_i)} \right\} \right]. \tag{45}$$

To complete the proof, we expand terms in the log-likelihood in (45):

$$\log p(\hat{\boldsymbol{\beta}} \mid \boldsymbol{\beta}, \boldsymbol{s}) = -\frac{1}{2} \sum_{i=1}^{n} \frac{(\beta_i - \hat{\beta}_i)^2}{s_i^2} + \text{const}, \tag{46}$$

where "const" is a placeholder for terms that do not involve $q$ (or $g$). Plugging this identity into (45), and with a bit of additional algebraic manipulation, we recover (40). $\square$

---

**Algorithm 1** cEBMF algorithm

---

**Require:** $n \times p$ data matrix, $\mathbf{Z}$; covariate or "side information" matrices, $\mathbf{X}$ $(n \times n_x)$ and $\mathbf{Y}$ $(p \times n_y)$; $K$, the number of factors; the prior families $\mathcal{G}_{\ell,k}$ and $\mathcal{G}_{f,k}$, $k = 1, \ldots, K$; and initial estimates of the first and second moments of $\mathbf{L}$ $(n \times K)$, $\mathbf{F}$ $(p \times K)$, which are denoted by $\bar{\mathbf{L}}, \bar{\mathbf{F}}, \bar{\mathbf{L}}^2, \bar{\mathbf{F}}^2$. Compute the expected residuals, $\bar{\mathbf{R}} = \mathbf{Z} - \bar{\mathbf{L}}\bar{\mathbf{F}}^T$.
  **repeat**
    Update the residual variances $\boldsymbol{\tau}$ using (35), (36) or (37).
    **for** $k = 1, \ldots, K$ **do**
      Remove the effect of the $k$th factor from the expected residuals, $\bar{\mathbf{R}}^k = \bar{\mathbf{R}} + \bar{\ell}_k \bar{\boldsymbol{f}}_k^T$.
      Perform a single-factor update for factor $k$ (Algorithm 2).
      Update the expected residuals, $\bar{\mathbf{R}} = \bar{\mathbf{R}}^k - \bar{\ell}_k \bar{\boldsymbol{f}}_k^T$.
    **end for**
  **until** some convergence criterion is met
  **return** $\bar{\mathbf{L}}, \bar{\mathbf{F}}, \boldsymbol{\tau}, g_1^{(\ell)}, \ldots, g_K^{(\ell)}, g_1^{(f)}, \ldots, g_K^{(f)}$

---

---

**Algorithm 2** cEBMF single-factor update

---

**Require:** covariate or "side information" matrices, $\mathbf{X}$ $(n \times n_x)$ and $\mathbf{Y}$ $(p \times n_y)$; $k \in \{1, \ldots, K\}$, the dimension to update; the prior families $\mathcal{G}_{\ell,k}$ and $\mathcal{G}_{f,k}$; an implementation of $\text{cEBNM}(\hat{\boldsymbol{\beta}}, \boldsymbol{s}, \mathbf{D}, \mathcal{G}) \to (\hat{\boldsymbol{\theta}}, \hat{q})$ (eq. 13) for prior families $\mathcal{G} = \mathcal{G}_{\ell,k}$ and $\mathcal{G} = \mathcal{G}_{f,k}$; the expected residuals, $\bar{\mathbf{R}}^k$; estimates of the second moments, $\bar{\mathbf{L}}^2$ $(n \times K)$, $\bar{\mathbf{F}}^2$ $(p \times K)$; and the residuals variances, $\boldsymbol{\tau}$.

1. $\hat{\boldsymbol{\beta}} \leftarrow \hat{\boldsymbol{\ell}}(\bar{\mathbf{R}}^k, \bar{\boldsymbol{f}}_k, \bar{\boldsymbol{f}}_k^2, \boldsymbol{\tau})$
2. $\boldsymbol{s} \leftarrow \boldsymbol{s}_\ell(\bar{\boldsymbol{f}}_k^2, \boldsymbol{\tau})$
3. $(g_k^{(\ell)}, q_k^\ell) \leftarrow \text{cEBNM}(\hat{\boldsymbol{\beta}}, \boldsymbol{s}, \mathbf{X}, \mathcal{G}_{\ell,k})$
4. Compute posterior moments $\bar{\ell}_{ik} := E_q[\ell_{ik}]$ and $\bar{\ell}_{ik}^2 := E_q[\ell_{ik}^2]$, $i = 1, \ldots, n$.
5. $\hat{\boldsymbol{\beta}} \leftarrow \hat{\boldsymbol{f}}(\bar{\mathbf{R}}^k, \bar{\ell}_k, \bar{\ell}_k^2, \boldsymbol{\tau})$
6. $\boldsymbol{s} \leftarrow \boldsymbol{s}_f(\bar{\ell}_k^2, \boldsymbol{\tau})$
7. $(g_k^{(f)}, q_k^f) \leftarrow \text{cEBNM}(\hat{\boldsymbol{\beta}}, \boldsymbol{s}, \mathbf{Y}, \mathcal{G}_{f,k})$
8. Compute posterior moments $\bar{f}_{jk} := E_q[f_{jk}]$ and $\bar{f}_{jk}^2 := E_q[f_{jk}^2]$, $j = 1, \ldots, p$.
9. **return** $\bar{\ell}_k, \bar{\ell}_k^2, \bar{\boldsymbol{f}}_k, \bar{\boldsymbol{f}}_k^2, g_k^{(\ell)}, g_k^{(f)}$

---

## A.4   Detailed algorithms

In summary, the cEBMF algorithm is a block co-ordinate ascent algorithm [69] for finding a local maximum of the ELBO (17), in which the "blocks"—i.e., the subsets of parameters to be updated—are the individual factors $k = 1, \ldots, K$ (Sec. A.1) and the residual variances $\boldsymbol{\tau}$ (Sec. A.2). This co-ordinate ascent algorithm is described in Algorithm 1, and the single-factor update is described in Algorithm 2. (And it is described informally in Sec. 3.3.2.) In practice, we run Algortithm 1 until the increase in the ELBO across two successive iterations is smaller than some specified tolerance, or until we have reached an upper bound on the number of iterations.

Two features of the empirical Bayes approach to matrix factorization discussed in [13] are worth highlighting here. First, there is a simple stepwise procedure for obtaining good initial estimates of $\mathbf{L}$ and $\mathbf{F}$ by introducing the factors sequentially. This was called a "greedy initialization" in [13]. Second, instead of fixing the number of factors, the EBMF approach can also select $K$ automatically by adapting the priors $g_k^{(\ell)}, g_k^{(f)}$ separately for each factor $k$. The idea is that factors that are not useful for explaining variation in the data should produce priors that are concentrated near zero (this feature of course requires that the chosen prior families $\mathcal{G}_{\ell,k}, \mathcal{G}_{f,k}$ include distributions that are concentrated near zero). Therefore, $K$ can initially be set to a large value, and the cEBMF algorithm will automatically determine an appropriate number of factors by "shrinking" the unneeded factors.

### A.4.1   Computational complexity

Since cEBMF is a modeling and algorithmic framework, and not a specific method or algorithm, we cannot give the exact computational complexity of Algorithm 1. However, we can provide some rules

of thumb. Steps 1, 2, 5 and 6 in Algorithm 2 (also, Steps 1 and 2 in Sec. 3.3.2) involve preparing the inputs for the cEBNM solver (13). Since these steps do not depend on the prior families $\mathcal{G}_{\ell,k}, \mathcal{G}_{f,k}$, we can give their computational complexity: when $\mathbf{Z}$ is a "dense" (non-sparse) matrix, the time complexity for updating a single factor $k$ is $O(np)$; when $\mathbf{Z}$ is sparse, the complexity is $O(S)$, where $S$ is the number of nonzero entries in $\mathbf{Z}$. (Note this requires careful implementation that avoids directly storing $\bar{\mathbf{R}}$). Steps 3 and 7 in Algorithm 2 (or Steps 3 and 4 in Sec. 3.3.2) will depend on the details of the cEBNM solver and the type of side information. However, when the priors on $\mathbf{L}, \mathbf{F}$ are simple and involve low-dimensional covariates, the other steps are expected to dominate, in which case the complexity of Algorithm 1 is expected to be $O(npK)$ or $O(SK)$.

# B  Details of the experiments

## B.1  Simulations

We simulated data sets from different cEBMF models. In all cases, the data were generated with homoskedastic noise, $\tau_{ij} = \tau$.

**Sparsity-driven covariate simulations.**  This simulation was intended to illustrate the behaviour of cEBMF when provided with simple row and column-covariates that inform only the sparsity of $\mathbf{L}$ and $\mathbf{F}$ (and not the magnitudes of their elements). The side information was stored in $1,000 \times 10$ and $200 \times 10$ matrices $\mathbf{X}$ and $\mathbf{Y}$, and the $1,000 \times 200$ matrix $\mathbf{Z}$ was simulated using a simple cEBMF model with $K = 2$ and with spike-and-slab priors chosen to ensure that 90% of the elements of $\mathbf{LF}^T$ were zero. Specifically, we used the following priors:

$$\begin{aligned}
\ell_{ik} &\sim \pi_{ik}\delta_0 + (1 - \pi_{ik})N(0, 1) \\
f_{jk} &\sim \alpha_{jk}\delta_0 + (1 - \alpha_{jk})N(0, 1) \\
\pi_{ik} &:= \phi(\boldsymbol{\theta}_k^T \boldsymbol{x}_i) \\
\alpha_{jk} &:= \phi(\boldsymbol{\omega}_k^T \boldsymbol{y}_j),
\end{aligned} \tag{47}$$

in which $\boldsymbol{\theta}_k$ and $\boldsymbol{\omega}_k$ were chosen to achieve 90% zeros in $\mathbf{LF}^T$.

**Uninformative covariate simulations.**  To verify that cEBMF was robust to situations in which the side information was not helpful, we considered an "uninformative covariate" setting in which the covariates were just noise. The data sets were simulated in the same way as in the sparsity-driven covariate simulations except that the true factors were simulated as $\ell_{ik} \sim \pi\delta_0 + (1 - \pi)N(0, 1)$, $f_{jk} \sim \alpha\delta_0 + (1 - \alpha)N(0, 1)$, with $\pi, \alpha$ chosen to achieve a target sparsity of 90% zeros in $\mathbf{LF}^T$.

**Tiled-clustering simulations.**  In this setting, we simulated rank-3 matrix factorizations in which $\mathbf{L}$—but not $\mathbf{F}$—depended on the 2-d locations of the data points. (One of these simulations is shown in Fig. 1.) This was accomplished as follows. First, we generated a periodic tiling of $[0, 1] \times [0, 1]$, randomly labeling each tile 1, 2 or 3. For each data point $i = 1, \ldots, 1,000$, we sampled its 2-d location uniformly from $[0, 1] \times [0, 1]$, then we set $\ell_{ik} = 1$ if the data point fell in the tile with label $k$, otherwise $\ell_{ik} = 0$. The $200 \times 3$ matrix $\mathbf{F}$ was simulated from a scale mixture of zero-centered normals that did not depend on tile membership.

**Shifted tiled-clustering simulations.**  To assess robustness to model misspecification, we simulated data using a prior that could not be recovered by the prior family we used in cEBMF. These simulations were the same as the tiled-clustering simulations except that we generated the $i$th row of $\mathbf{L}$ as follows: $(1, 2, 3)$ if data point was $i$ in the tile with label 1; $(3, 1, 2)$ if data point was in the tile with label 2, and $(2, 3, 1)$ if data point was in the tile with label 3.

## B.2  Additional details on the methods compared

We first describe how the methods were run on the simulated data sets. Modifications to the methods for the MovieLens and spatial transcriptomics data are given in the main text, with additional technical details below. For all methods, when possible to do so, we set the rank, $K$, to match the rank of the simulated matrix factorization.

For EBMF and cEBMF, we assumed homoskedastic noise ($\tau_{ij} = \tau$), and the prior families were chosen to align with how the data were simulated (except for the shifted-tiled clustering simulations, which were intended to illustrate the methods' behaviour when the priors were misspecified). For EBMF, the prior families were all elaborations of the "spike and slab" priors (Sec. 3.2.1). For cEBMF, the priors were of the same form as EBMF in which the mixture weights were parameterized using either a multinomial regression (i.e., a single-layer neural network with a softmax link function) or a multilayer perceptron.

In the sparsity-driven covariate and uninformative covariate simulations, the priors for $\mathbf{L}$ and $\mathbf{F}$ in EBMF were all scale mixtures of normals with a fixed grid of scales [50, 70]. For cEBMF, we used priors of the same form, except that side information was incorporated into the prior mixture weights as follows using priors of the following form:

$$g(\boldsymbol{d}_i, \boldsymbol{\theta}) = \pi_0(\boldsymbol{d}_i, \boldsymbol{\theta})\delta_0 + \sum_{m=1}^{M} \pi_m(\boldsymbol{d}_i, \boldsymbol{\theta})N(0, \sigma_m^2). \tag{48}$$

The mixture weights $\pi_0, \ldots, \pi_M$ were implemented using a standard multinomial regression model with the softmax link function.

In the tiled-clustering and shifted tiled-clustering simulations, the true $\mathbf{L}$ was always non-negative. Therefore, we chose the prior families in EBMF and cEBMF to produce *semi-non-negative matrix factorizations* [16] with a non-negative $\mathbf{L}$. Specifically, we assigned mixture-of-exponentials priors to $\mathbf{L}$, similar to the scale-mixture-of-normals priors, except with support for non-negative numbers only [35]. And we assigned the scale-mixture-of-normal priors, same as above, to $\mathbf{F}$. In cEBMF, side information was incorporated into the mixture weights in the prior in a manner similar to above:

$$g(\boldsymbol{d}_i, \boldsymbol{\theta}) = \pi_0(\boldsymbol{d}_i, \boldsymbol{\theta})\delta_0 + \sum_{m=1}^{M} \pi_m(\boldsymbol{d}_i, \boldsymbol{\theta})\exp(\lambda_m), \tag{49}$$

in which $\exp(\lambda)$ denotes the exponential distribution with scale parameter $\lambda$, and $\lambda_{m-1} < \lambda_m$, $m = 2, \ldots, M$. As before, the mixture weights $\pi_0, \ldots, \pi_M$ were implemented using a standard multinomial regression model with the softmax link function.

The deep learning methods (VAE, cVAE, NCF) were all implemented in PyTorch. All the models were trained for 50 epochs using the Adam optimizer with learning rate 0.001 and batch size 64. VAE had three hidden layers (of width 128, 64 and 30) in both the encoder and decoder (20 hidden dimensions). ReLU activations were used throughout. We use the ELBO from [26] to train the model. We proceeded similarly for the cVAE, conditioning both the encoder and decoder on the available covariate data $\mathbf{X}, \mathbf{Y}$. For cVAE, we used the training objective from [27]. NCF models $\mathbf{Z}$ using two separate multilayer perceptrons for the row and column covariates [28]. The multilayer perceptrons were implemented in a similar way to the VAE encoders and decoders; that is, three hidden layers (of width 128, 64 and 30) with RELU activations. The penalty parameters in PMD were tuned via cross-validation as recommended by the authors. SSVD (R package "ssvd") was run with its default values.

For the spatial transcriptomics data, we fit cEBMF and EBMF using gene-specific residual variances, $\sigma_{ij}^2 = \sigma_j^2$. We used mixture-of-exponential priors for $\mathbf{F}$, and the parameterized mixture-of-exponential priors (49) for $\mathbf{L}$ in which the mixture weights were learned using a multilayer perceptron instead of a multinomial regression. The multilayer perceptions were defined as sequential models with a dense layer with 64 units and ReLU activations. We use two subsequent dense layers, each with 64 units, and ReLU activations using an L2 regularization coefficient of 0.001 to prevent overfitting. These regularized layers were followed by a dropout layer (with a dropout rate of 0.5). The subsequent layers were four dense layers each with 64 units, ReLU activations and L2 regularization coefficient of 0.001. The final layer was a dense layer with a softmax activation. These models were trained during each single-factor update using 300 epochs and a batch size of 1,500.

In the simulations, cEBMF was implemented in R, in which learning the parameterized priors was performed using the Keras R interface [71] to TensorFlow [72]. For the MovieLens and spatial transcriptomics data sets, we used the PyTorch-based implementation of cEBMF which we have made available as a Python package on GitHub.

# C  Additional results

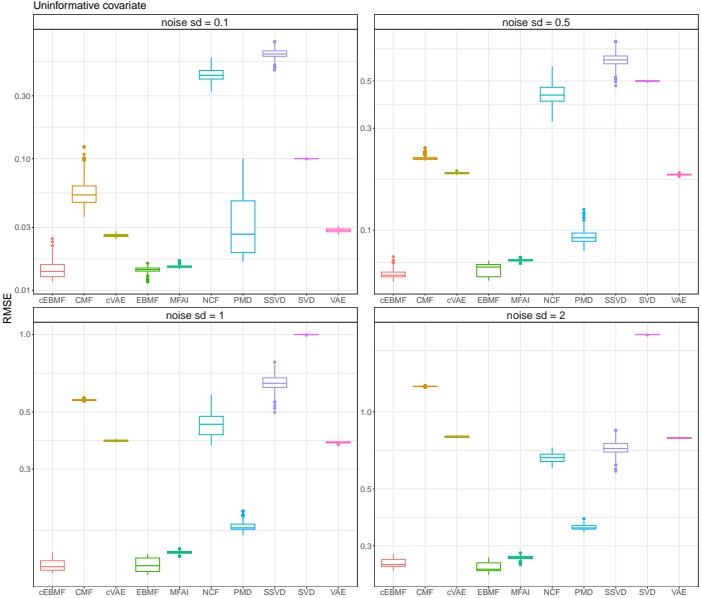

Figure 5: Simulation results from the "uninformative covariate" setting in which the data were simulated under different noise levels, $\tau$. Note that for improved visualization the RMSE is shown on the log-scale.

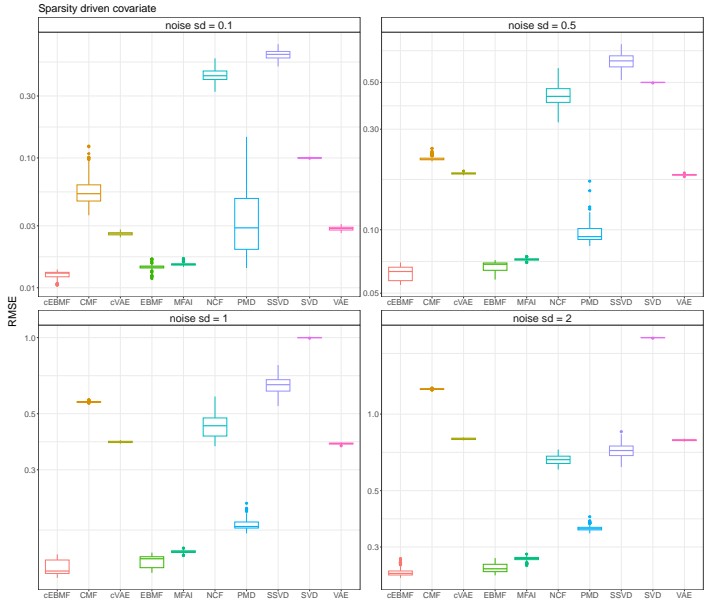

Figure 6: Simulation results from the "sparsity-driven covariate" setting in which the data were simulated under different noise levels, $\tau$. Note that for improved visualization the RMSE is shown on the log-scale.

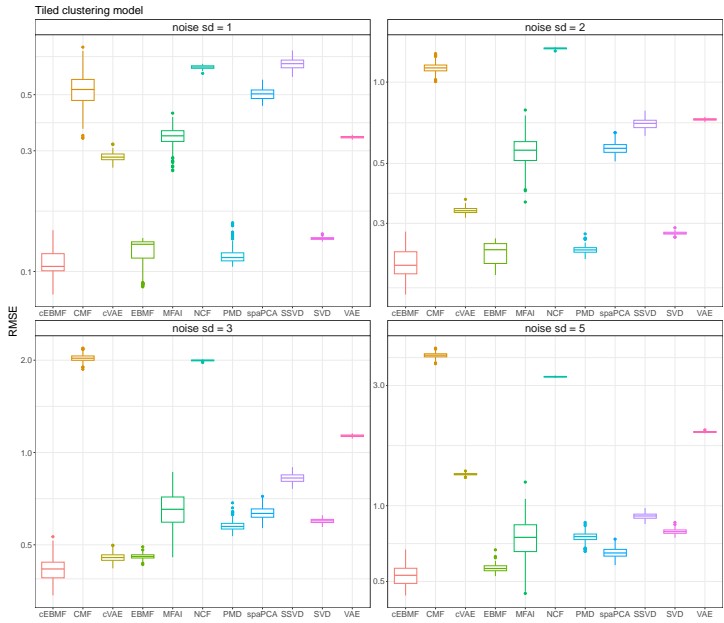

Figure 7: Simulation results from the "tiled-clustering" setting in which the data were simulated under different noise levels, $\tau$. Note that for improved visualization the RMSE is shown on the log-scale. (spaPCA = Spatial PCA)

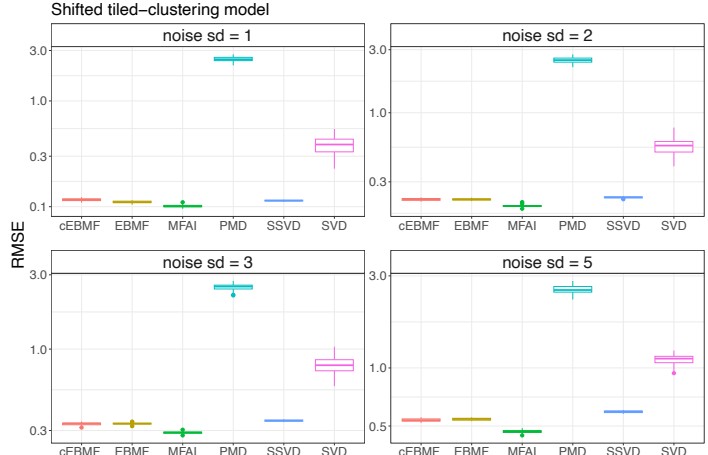

Figure 8: Simulation results from the "shifted tiled-clustering" setting in which the data were simulated under different noise levels, $\tau$. Note that for improved visualization the RMSE is shown on the log-scale.

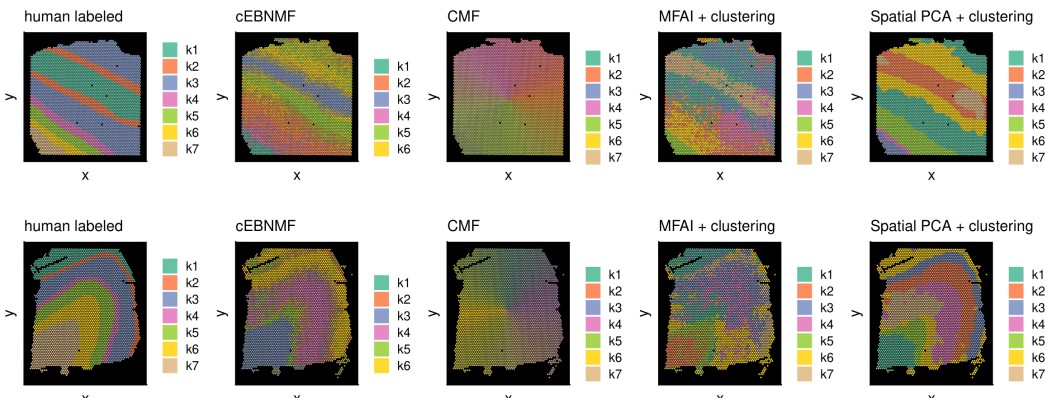

Figure 9: Additional results on slides 4 (top) and 10 (bottom) of the DLPFC spatial transcriptomics data. See Fig. 4 for additional information about these results.

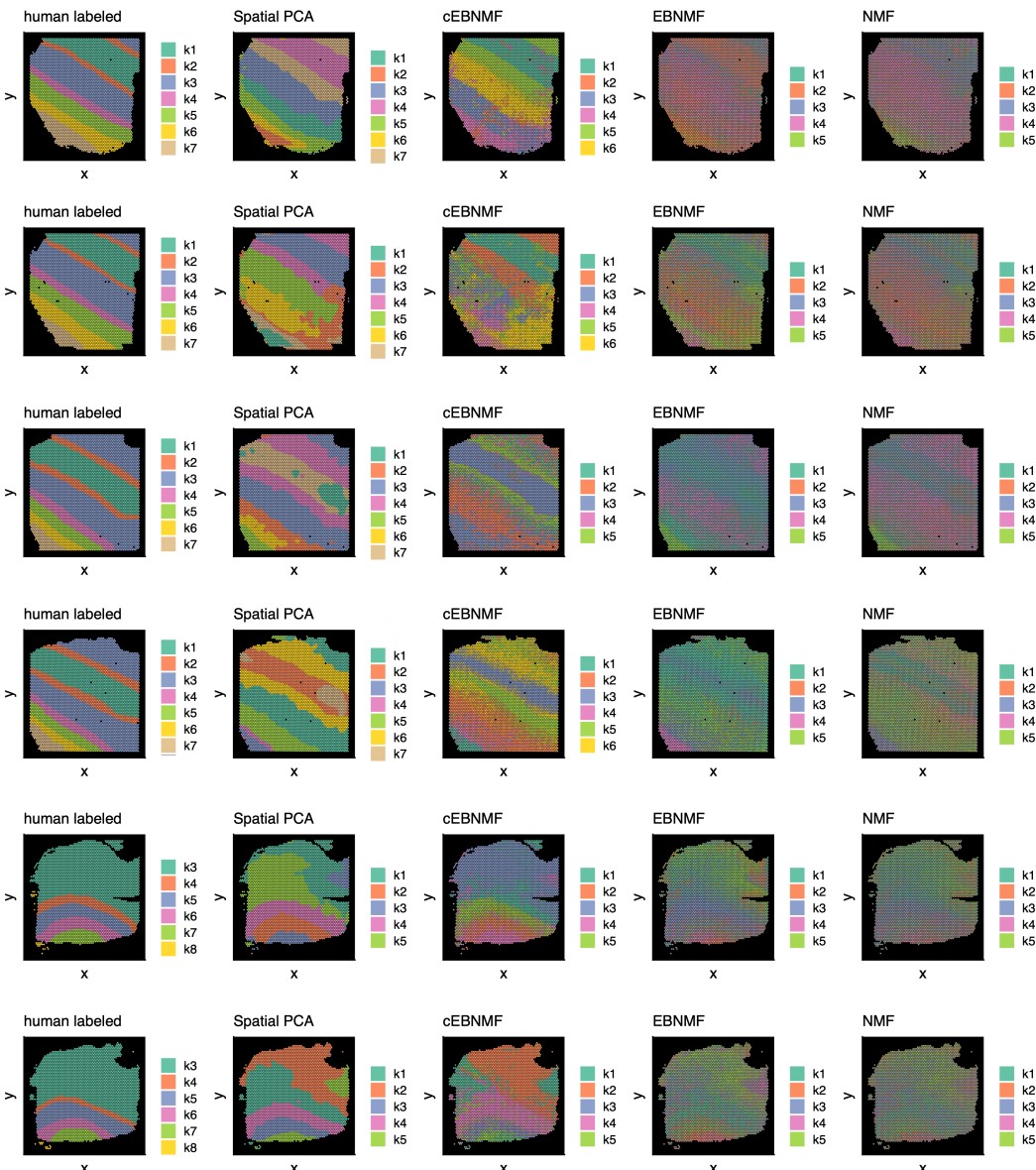

Figure 10: Selected results on slices 1 (top row) through 6 (bottom row) of the DLPFC spatial transcriptomics data.

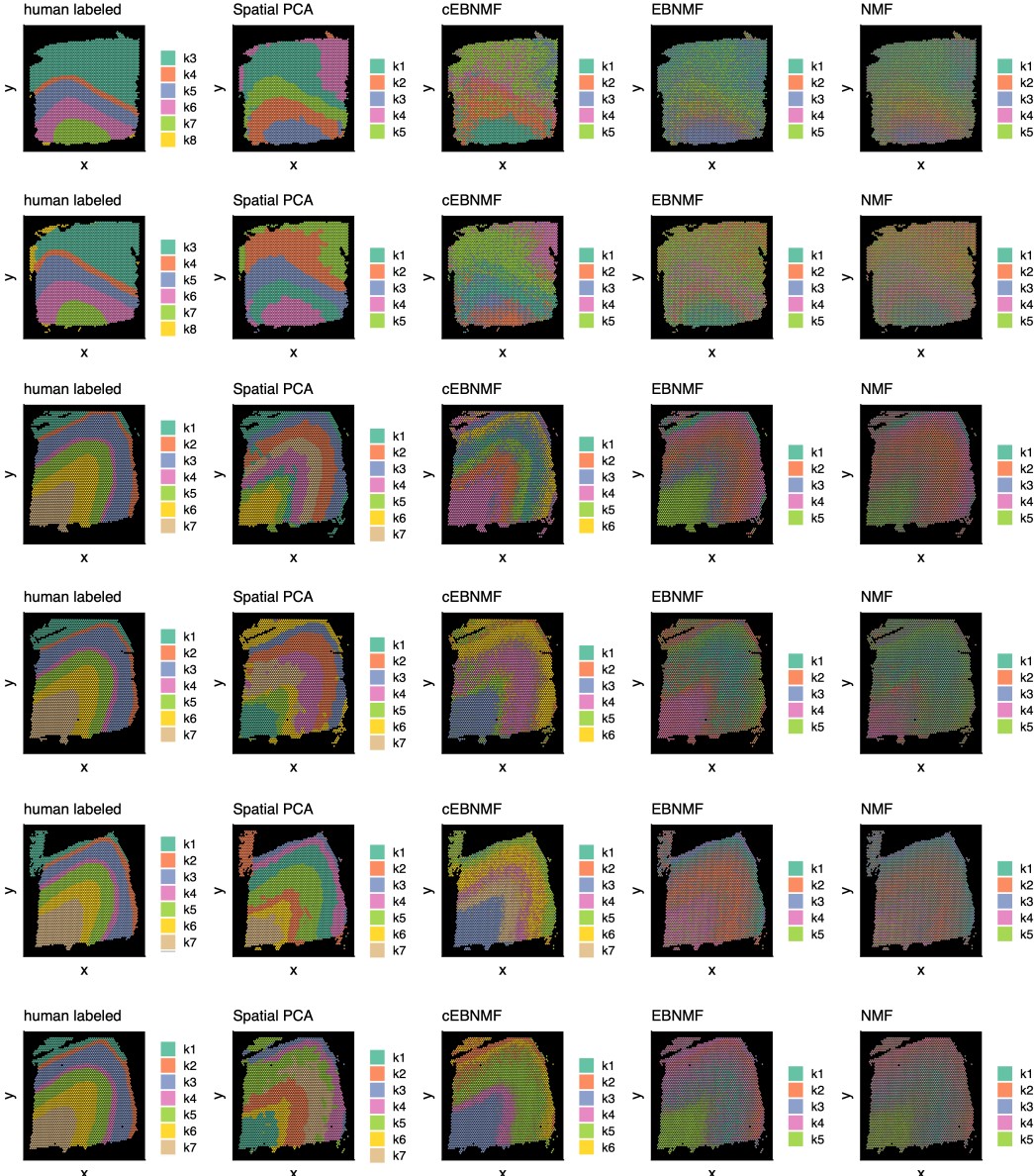

Figure 11: Selected results on slices 7 (top row) through 12 (bottom row) of the DLPFC spatial transcriptomics data.

