# OpenReview forum: "Covariate-moderated Empirical Bayes Matrix Factorization"
_NeurIPS.cc/2025/Conference — NeurIPS 2025 poster_

### Official Review · Reviewer_KUPC · 2025-06-27

**Clarity:** 3
**Significance:** 2
**Originality:** 3
**Rating:** 3
**Confidence:** 3

**Summary:**

This paper proposes Covariate-moderated Empirical Bayes Matrix Factorization (cEBMF), a general framework that extends EBMF by allowing covariate-dependent priors on latent factors. The key idea is to learn a mapping from covariates to priors via flexible models (e.g., neural networks), enabling the integration of high-dimensional side information. Empirical results demonstrate strong performance across synthetic and real-world datasets, with clear gains over competing methods and favorable scalability.

**Questions:**

1. Add experimental validation to demonstrate that the proposed method can leverage more complex side information, such as incorporating semantic and visual information of items in the task of collaborative filtering.
2. Add discussion on why an end-to-end neural network is not directly used to model all types of information and accomplish the target task.
3. Add some new baselines about deep learning.
4. Add the discussion of the tasks that Matrix Factorization (MF) are suitable for.

**Ethical Concerns:**

["NO or VERY MINOR ethics concerns only"]

**Final Justification:**

My concerns still remain in spite of the author reponse. So I would keep my original score of borderline reject.

**Limitations:**

Even with the introduction of side information, matrix factorization (MF) remains a linear model, which seems to limit its applicability.

**Quality:**

3

**Strengths And Weaknesses:**

Strengths
1. This paper introduces cEBMF, a general framework integrating any side information (text, images, graphs) via covariate-dependent priors. This surpasses existing methods limited to specific data types (e.g., spatial coordinates).
2. This paper allows plug-and-play priors, enabling adaptation to diverse problems.
3. The proposed method is supported by solid theoretical foundations.

Weaknesses
1. This method can be regarded as a two-stage approach: it first uses a neural network to process side information, treating it as a prior, and then performs matrix factorization. I think the paper lacks sufficient discussion on why an end-to-end neural network is not directly used to model all types of information and accomplish the target task.
2. The deep learning baselines (e.g., VAE, NCF) are relatively simple and may not fully represent the state of the art. The paper would benefit from a more nuanced positioning.
3. Lacks analysis of the experiment results — why the proposed linear model outperforms nonlinear models in recommendation tasks.

---

> ### Author Rebuttal · Authors · 2025-07-29
>
> We thank the reviewer for the time and the detailed review of our work.
>
> We agree on the fact that the semantic and visual information of items was not used in the collaborative filtering task. Collaborative filtering is not our main field. Could the reviewer suggest some good benchmarks that have this information?
>
> The reviewer wrote, **"This method can be regarded as a two-stage approach: it first uses a neural network to process side information, treating it as a prior, and then performs matrix factorization. I think the paper lacks sufficient discussion on why an end-to-end neural network is not directly used to model all types of information and accomplish the target task. "**
>
> The reviewer might be unfamiliar with Empirical Bayes procedures, but this two-step procedure is the very essence of Empirical Bayes (EB) (we recommend Efron's book on Large-scale inference for an introduction). The nice aspect of EB is that it doesn't require the user to specify some hyperparameters prior to the analysis. Further, the reviewer may have missed that the prior model (here neural nets) is retrained whenever we are fitting a new factor, and the overall procedure is not quite as described by the reviewer.
>
> The reviewer wrote **"Lacks analysis of the experiment results — why the proposed linear model outperforms nonlinear models in recommendation tasks."**
>
> We agree that it may at first glance look surprising, but there are several instances of real data examples in which linear models are ahead of non-linear models. For example, in genetics, polygenic risk score methods (which are linear models) are still much better than the nonlinear/ML-based methods despite having millions of data points.
>
> We hope that we have answered your questions and that you may consider increasing our rating.

---

### Official Review · Reviewer_6qut · 2025-07-02

**Clarity:** 2
**Significance:** 3
**Originality:** 3
**Rating:** 4
**Confidence:** 4

**Summary:**

The paper introduces cEBMF, a flexible and modular framework for matrix factorization. The method integrates various types of "side information" by incorporating them into adaptive, covariate-dependent prior distributions. cEBMF learns these priors directly from the data, enhancing the inference and summarization of multivariate data structures. The benefits of cEBMF are demonstrated through simulations and applications to spatial transcriptomics and MovieLens datasets.

**Questions:**

### Choice of $K$

The cEBMF framework demonstrates impressive flexibility in incorporating various prior assumptions and types of side information. However, the rank $K$ is treated as a fixed hyperparameter in the current experimental setup.

Given the modular nature of cEBMF and its stated ability to accommodate different prior families, have the authors considered extending the framework with non-parametric Bayesian approaches, such as Dirichlet Process priors, to allow for automatic inference of the optimal rank $K$ ? What are the theoretical or practical challenges in implementing such an extension, and are there plans for future work in this direction?

### Limited sample scenario with large $K$

A common challenge in matrix factorization models is dealing with scenarios where the number of latent factors, $K$, is relatively large compared to the available data size.

What is the expected behavior and potential limitations of cEBMF in such "small data, large $K$" regimes? Specifically, since training the function $\pi(\cdot, \cdot)$ (which determines the covariate-dependent prior probabilities) in a "small data size, large K" regime seems to present significant challenges despite its flexibility, how might the empirical Bayes nature and the covariate moderation mechanisms of cEBMF help to mitigate issues like overfitting or unstable parameter estimation in these challenging settings?

### Optimization

The cEBMF learning algorithm relies on an iterative coordinate ascent approach to maximize the ELBO. The performance and convergence of such optimization procedures can often be sensitive to the initial values of the latent factors and prior parameters.

Could the authors provide more details on the initialization strategies employed for the latent matrices ($L$ and $F$) and the parameters of the covariate-moderated priors ($\theta$​)? How do these choices impact the final results and convergence speed, particularly in the challenging scenarios presented in the experiments? Does the framework incorporate any specific robust initialization techniques or strategies to mitigate potential issues arising from poor initialization?

**Ethical Concerns:**

["NO or VERY MINOR ethics concerns only"]

**Final Justification:**

As per the system’s instructions, paste my extended, prior final justification:

I am satisfied with the authors’ responses provided during the rebuttal period. I have also reviewed the questions, comments, and exchanges between the other reviewers and the authors. Several noteworthy points were raised, and in my view, the majority of the authors’ responses were appropriate and well-reasoned. I therefore maintain that my original assessment remains valid.

**Limitations:**

no issue

**Paper Formatting Concerns:**

no issue

**Quality:**

3

**Strengths And Weaknesses:**

### Strengths

- cEMBF's modular and flexible design is attractive and the novelty of incorporating any type of side information is practical. In addition, the proposed algorithm tries to learn the covariate-dependent prior distribution representing the side information is inspiring.

- The cEMBF framework generalizes different matrix factorization algorithms with different regularizations under the view of covariate-dependent priors. It is simple yet inspiring to see well-established algorithms like sparse NMF can also be viewed under this framework. The scalability of the framework is also promising.

### Weaknesses

- The performance of cEMBF on small dataset, large $K$ regime is unclear, yet it is a very common scenario in real application
- The choice of the latent dimension is $K$ for all experimental setup, while given the flexibility claimed in cEMBF, one should be able to incorporate Dirichlet process prior so $K$ is learnable.
- While the performance of many ELBO optimization procedures can indeed be sensitive to the choice of initialization, the provided paper content does not explicitly detail the specific initialization strategies used for the latent matrices (L and F) or the prior parameters $\theta$.

---

> ### Author Rebuttal · Authors · 2025-07-29
>
> We thank the reviewer for the time and the detailed review of our work.
>
> **Regarding the choice of K**
> This is a very good and important point. Due to the page limit we discuss the choice of K in the appendix (A.4). We do not consider the Dirichlet Process Mixture but rather a more classical and simpler approach from the Empirical Bayes literature.
>
> We write in the appendix (A.4)  "A noteworthy aspect of empirical Bayes methods for matrix factorization, highlighted in Bishop 1999, is their inherent ability to automatically determine $K$ . This feature comes from the fact that the maximum-likelihood estimates for $g_k^{(\ell)}$ and $g_k^{(f)}$ can converge to a point-mass at zero (assuming that the families $G_{\ell, k}$ and $G_{f, k}$ include a point-mass at zero). Thus, if $K$ is initially set to a large value, certain row or column factors will converge to zero, which effectively "zeroes out" the corresponding
> component $k$.
>
> So, as long as the prior model family contains the point mass at 0, then cEBMF automatically estimates K. This means that if K is set sufficiently large, then some loading/factor combinations will be optimized to be exactly 0.
>
> **Regarding the choice of prior**
>
> Certainly, other priors could be used and may be more appropriate for different data sets or for different types of side information. We have not considered the use of hierarchical Dirichlet models nor multi-kernel Gaussian processes, and it is hard to comment on the benefits of such approaches without being more specific about the type of data or side information (categorical, spatial, temporal, etc) being considered. It is straightforward to use cEBMF with hierarchical Dirichlet models or multi-kernel Gaussian processes prior, given that you can solve the cEBNM mapping for these priors. So, more generally, cEBMF can be used with any prior model in which the cEBNM mapping can be computed or at least numerically approximated (see equation 13).
>
> **Regarding the initialization**
>
> This is good point raised by the reviewer. For unconstrained factorization, we simply start by performing a truncated SVD of the data matrix and using the U and V  matrices as initialization for L and F. For non-negative matrix factorization and semi-non-negative factorization, we use a fast NMF routine (in R or in Python) to provide the initialization for L and F. We found that when using cEBMF for non-negative matrix factorization, using fast NMF routines for the initial starting point for L and F helps compared to using SVD initialization.
>
> For the covariate-moderated priors model, we have not experimented with specific initialization, but for future work, we aim at using a pretrained model for specific applications (e.g., a model for cell recognition), and we hope to obtain some speedup as well as performance improvement from those.
>
> We will add those details in the appendix in the final version.
>
> The reviewer wrote: **"A common challenge in matrix factorization models is dealing with scenarios where the number of latent factors is relatively large compared to the available data size".**
>
> We think that, in general, the main goal is to find a lower space representation of a relatively large data matrix. So, usually k is small compared with the data size. For example, in single-cell applications, the number of genes is >10000, the number of cells is >10000, but k is of the order 100.
>
> We hope that we have answered your questions and that you may consider increasing our rating.

---

> > ### Comment · Reviewer_6qut · 2025-08-06
> >
> > Thank you for the detailed rebuttal. The clarifications on initialization and the modularity of cEBMF with different priors are appreciated.
> > Regarding the treatment of rank $K$, I remain concerned. While empirical Bayes shrinkage with point-mass-at-zero priors can theoretically zero out unused components, its practical effectiveness depends critically on prior strength. If $K$ is set large and the prior is weak, is there a risk of retaining noise as spurious factors? Have the authors evaluated the sensitivity of this mechanism to prior choice or considered diagnostics to detect when rank overestimation occurs?
> >
> > While many applications involve large datasets with small $K$ small-sample or low-signal regimes are also common in practice. Discussion or evaluation in such settings would help better assess the robustness of the method.

---

> > > ### Author Response · Authors · 2025-08-07
> > >
> > > Thank you for the constructive and insightful feedback.
> > >
> > > To correctly estimate K, it is essential that the family of priors optimized by the model is adaptive—this is a foundational principle in empirical Bayes. We refer the reviewer to the influential paper “False discovery rates: a new deal” (Stephens, Biostatistics 2019), where one major theme is adapting prior shrinkage to match the signal strength in the data. This directly relates to the cEBMF ability of selecting K since we are using similar priors.
> > >
> > > In practice, following suggestions from Stephens (see supplement), we add a small additional weight to the point mass during optimization, which is equivalent to placing a Dirichlet prior on the mixture weight. Roughly, this is analogous to adding around 10 exactly-zero observations. We will clarify this detail in our supplement.
> > >
> > > As discussed by Ignatiadis & Wager (“Confidence Intervals for Nonparametric Empirical Bayes Analysis,” JASA 2022), this extra regularization improves numerical stability and encourages the estimated prior to load onto the spike-at-zero component when appropriate.
> > >
> > > Such spike-and-slab or sparse priors have been shown to facilitate robust, data-driven selection of K; see, for example, "Sparse Bayesian Factor Analysis When the Number of Factors Is Unknown" (Frühwirth-Schnatter et al., Bayesian Analysis 2025).
> > >
> > > We appreciate the reviewer’s input and will clarify this adaptive regularization in our supplementary materials.

---

> > > > ### Comment · Reviewer_6qut · 2025-08-07
> > > >
> > > > Thank you for your clarifications and for pointing me to the relevant papers, which are helpful. In light of your comments, I believe my original assessment remains appropriate.

---

### Official Review · Reviewer_cne9 · 2025-07-03

**Clarity:** 3
**Significance:** 3
**Originality:** 3
**Rating:** 5
**Confidence:** 3

**Summary:**

This paper introduces a covariate-moderated empirical Bayes matrix factorization framework (cEBMF) for incorporating side information into matrix factorization. The authors first extend the EBMF model by allowing the priors over latent factor entries to depend on covariates (side information), enabling the model to adaptively learn sparsity or structure guided by auxiliary data. Then, a modular variational algorithm is proposed, where matrix factorization is learned by solving covariate-modulated empirical Bayes normal means (cEBNM) subproblems iteratively. The framework supports various prior families, including spike-and-slab for sparsity or non-negativity. Experiments on simulated data, MovieLens 100K data, and spatial transcriptomics data show that cEBMF achieves superior performance and scalability compared to existing methods like EBMF, MFAI, Spatial PCA, and deep learning baselines.

**Questions:**

i) While the authors included comparisons between cEBMF and EBMF, would it be possible to include an ablation study exploring different prior families? Additional visualizations showing performance variation under different priors would help.
ii) How robust is cEBMF to noisy or misspecified covariates? A sensitivity analysis or simulation study on covariate quality would be valuable.
iii) Have the authors considered methods for automatically inferring the number of latent components (K)? For example, a Bayesian nonparametric approach using a Dirichlet Process Mixture or similar mechanism might offer a flexible alternative.

**Ethical Concerns:**

["NO or VERY MINOR ethics concerns only"]

**Final Justification:**

I am satisfied about authors' response during the rebuttal period and appreciate their effort in trying to address my concerns in a short period of time, especially to my last point.
Originally I already leaned to accept their paper so I would keep my rating.

**Limitations:**

Yes.

**Paper Formatting Concerns:**

Please correct the citation formatting to align with the required style guidelines.

**Quality:**

3

**Strengths And Weaknesses:**

Strengths
i) The paper presents a well-founded theoretical framework with thorough derivations and proofs. The proposed algorithms are clearly described and easy to follow.
ii) Visualizations are informative, with multiple figures that effectively illustrate the experimental results. Applications to MovieLens 100K and spatial transcriptomics are particularly compelling.
iii) The incorporation of covariates (side information) into the factor model is innovative, and the extension of EBMF is both well-motivated and systematically developed.

Weaknesses
i) The paper lacks an ablation study evaluating the impact of covariates or the choice of different prior families on model performance.
i)) The distinction between “EBMF” and “EBNMF” is somewhat unclear and may be confusing for readers unfamiliar with the naming convention.

---

> ### Author Rebuttal · Authors · 2025-07-29
>
> We thank the reviewer for the time and the detailed review of our work.
>
> **The reviewer asks if it would be possible to include an ablation study exploring different prior families**?
>
> Regarding your question on the choice of the prior model, we perform a simulation study in which the prior model cannot recover the data-generative model (see shifted tiled clustering, section 4). We show that the results are very similar to the EBMF ones, and so using a completely misspecified covariate model has rather limited impact on the cEBMF performance.  See Figure 2 most right panel
>
> Further, the reviewer suggests that **"A sensitivity analysis or simulation study on covariate quality would be valuable."**
>
> We included a simulation study in which we use uninformative covariates (see Figure 2 most left panel). When the covariates are uninformative, cEBMF has similar performance to EBMF, which is a sign of the good behaviour of the fitting procedure, as the EBMF model corresponds to a cEBMF model without covariates.
>
> Lastly, the reviewer asks: **"Have the authors considered methods for automatically inferring the number of latent components (K)? For example, a Bayesian nonparametric approach using a Dirichlet Process Mixture or similar mechanism might offer a flexible alternative."**
>
> This is a very good and important point. Due to the page limit, we discuss the choice of K in the appendix (A.4). We do not consider the Dirichlet Process Mixture but rather a more classical and simpler approach from the Empirical Bayes literature.
> We write in appendix (A.4)
>
> "A noteworthy aspect of empirical Bayes methods for matrix factorization,
> highlighted in Bishop 1999, is their inherent
> ability to automatically determine $K$ . This
> feature comes from the fact that the maximum-likelihood estimates for
> $g_k^{(\ell)}$ and $g_k^{(f)}$ can converge to a point-mass at zero
> (assuming that the families $G_{\ell, k}$ and
> $G_{f, k}$ include a point-mass at zero). Thus, if $K$ is
> initially set to a large value, certain row or column factors will
> converge to zero, which effectively "zeroes out" the corresponding
> component $k$.
>
> So as long as the prior model family contains  the point mass at 0, then cEBMF automatically estimates K This means that if K is set sufficiently large, then some loading/factor combinations will be optimized to be exactly 0.
>
> We hope that we have answered your questions and that you may consider increasing our rating.

---

> > ### Comment · Reviewer_cne9 · 2025-08-06
> >
> > I appreaciate the rebuttal and efforts. The authors address many of my concerns and make it more clear. The authors' clarification about the choice of K that "zeroes out" effect with large K value makes sense to me. As I stated in my original review, this paper is technically solid and I will maintain my scores.

---

### Official Review · Reviewer_XAok · 2025-07-06

**Clarity:** 3
**Significance:** 2
**Originality:** 3
**Rating:** 4
**Confidence:** 4

**Summary:**

This paper introduces cEBMF, a framework that integrates side-information into Empirical Bayes Matrix Factorization by learning both spike-and-slab mixing weights and slab parameters as differentiable functions of covariates.  The authors demonstrate improved reconstruction accuracy over EBMF, MFAI, Spatial PCA, and VAE-based baselines on simulated data, MovieLens 100K, and spatial transcriptomics, and show scalability to up to one million rows.

**Questions:**

* How about the robustness to prior mis-specification? I suggest to include an ablation showing performance when the true slab distribution differs from the chosen parametric family.

* The paper instantiates the covariate-modulated prior using a specific parametric family. Can the authors characterize the representational capacity of that family—i.e. what classes of true slab distributions it can approximate well? Moreover, what would it take to replace this with a fully nonparametric density estimator (e.g. a normalizing flow or kernel density network)? I recognize that this adds optimization and identifiability challenges, but such an analysis could clarify the upper bound on prior flexibility and guide future nonparametric extensions.

**Ethical Concerns:**

["NO or VERY MINOR ethics concerns only"]

**Final Justification:**

The authors’ responses have addressed some of my concerns. However, the lack of comparison with methods from other families remains unresolved. Therefore, I will keep my original rating of 4

**Limitations:**

The author does not describe the limitation in the main part of the paper but mentioned several in the checklist.

**Quality:**

2

**Strengths And Weaknesses:**

## Strengths

* New Prior Modeling: developed a new covariate→prior-parameter mappings in an EBMF framework;

* Modular Learning  Structure: Decomposes the overall inference into repeated calls to a cEBNM solver, enabling plug-and-play of different solver backends.

* Empirical Validation across Domains: Provides thorough experiments on synthetic scenarios, recommendation data, and spatial omics, demonstrating consistent gains when side information is informative, and no degradation when it is not.

## Weakness

* By design, cEBMF cannot capture nonlinear interactions or higher-order effects; no comparison is made to deep nonlinear baselines (e.g. self-attention recommenders).

* Omits comparisons to modern graph-based or attention-based methods (e.g. GC-MC, LightGCN, SASRec, BERT4Rec), which also integrate side information.

* Requires pre-specifying a family for g_1 (Gaussian, Exponential, mixtures). When the chosen family mismatches data, performance may suffer; robustness to mis-specification is not studied.

* No formal analysis of identifiability, convergence rates, or generalization bounds—particularly important when covariates weakly correlate with latent factors.

---

> ### Author Rebuttal · Authors · 2025-07-29
>
> We thank the reviewer for the time and the detailed review of our work.
>
>
> **How the  method handles prior mis-specification?**
>
> This is an important point raised by the reviewer. We would like to highlight that we actually did what the reviewer is wondering. We performed a simulation study in which the covariate model cannot recover the data-generative model (shifted tiled clustering). We show that the results are very similar to the EBMF ones and so using a completely misspecified covariate model has rather limited impact on the cEBMF performance. See Figure 2 most right panel
> Regarding your questions about the choice of prior family and the use of mixture priors, the "spike-and-slab" prior presented in Sec. 3.2.1 is intended only to illustrate the key ideas. (Although, to be clear, the "spike-and-slab" prior is an example of a mixture prior—it is a mixture prior with two mixture components.)
>
> **Other questions**
>
>  For the simulations and spatial transcriptomics data application, we used slightly more flexible mixture priors; the details are given in Appendix C of the paper. Certainly, other priors could be used and may be more appropriate for different data sets or for different types of side information. We have not considered the use of hierarchical Dirichlet models nor multi-kernel Gaussian processes, and it is hard to comment on the benefits of such approaches without being more specific about the type of data or side information (categorical, spatial, temporal, etc) being considered. It is straightforward to use cEBMF with hierarchical Dirichlet models or multi-kernel Gaussian processes prior, given that you can solve the cEBNM mapping for these priors. So, more generally, cEBMF can be used with any prior model in which the cEBNM mapping can be computed or at least numerically approximated (see Equation 13).
>
> **Limitations**
>
> Regarding the limitations acknowledgment, while we agree that we don't have a specific section about it, we spent a great deal of time describing the limited scope of the method in the first section of our manuscript.
>
>
> We hope that we have answered your questions and that you may consider increasing our rating.

---

> > ### Comment · Reviewer_XAok · 2025-08-07
> >
> > Thank you to the authors for their response. Comparisons and analyses with modern graph-based or attention-based methods are important for a comprehensive assessment of this method’s contribution to the field. Therefore, I would like to maintain my original rating.

---

### Official Review · Reviewer_NLUp · 2025-07-13

**Clarity:** 3
**Significance:** 3
**Originality:** 3
**Rating:** 4
**Confidence:** 4

**Summary:**

The paper proposes cEBMF, a modular empirical‑Bayes matrix‑factorization framework that can ingest side information (text, images, graphs, coordinates) via covariate‑dependent prior families.
The framework unifies and generalizes prior methods (EBMF, NMF, Spatial PCA, MFAI, CMF) as special cases through flexible prior families and automatic hyper‑parameter learning.
The authors derive a simple coordinate‑ascent algorithm that reduces the learning problem to repeated covariate‑moderated empirical‑Bayes normal‑means sub‑problems optimizing a variational approximation, enabling plug‑and‑play priors and scalability.
In experiments, state‑of‑the‑art accuracy is shown across simulations, collaborative filtering (MovieLens 100K) and spatial transcriptomics (human DLPFC), while outperforming EBMF, MFAI, Spatial PCA, PMD, VAE/cVAE, NCF.
An open‑source R/Python implementation is provided including runtime benchmarks where cEBMF scales markedly better than covariate-dependent alternatives MFAI and Spatial PCA.

**Questions:**

How sensitive is the performance to the choice of covariate model?
What guarantees (if any) do you have about convergence or posterior consistency in this setting?
Is your method applicable in settings where covariates are high-dimensional or partially missing?

**Ethical Concerns:**

["NO or VERY MINOR ethics concerns only"]

**Final Justification:**

Thank you for the response and clarifications. In the light of the comments, I deem my original assessment appropriate.

**Limitations:**

No Limitations Section is provided. Arguably, some limitations are implicitly addressed, by describing the limited scope of the methods.

**Paper Formatting Concerns:**

No formatting concerns.

**Quality:**

3

**Strengths And Weaknesses:**

(+) generalizes many existing matrix factorization methods as special cases
(+) a computationally efficient implementation is provided
(+) The paper is written with generality in mind
(+) The experimental evaluation is extensive
(-) due to generality it was somewhat work for me as a reader to figure out, which priors and models have actually been used. Sometimes instantiating special cases helps clarity.
(-) compared to EBMF, the covariate-dependent cEBMF seems like an obvious extension of known pieces. One might even argue that it is a special case of EBMF where the prior family is parametrized using covariates.
(-) no theoretical results are provided beyond derivations of the algorithm and methods.

---

> ### Author Rebuttal · Authors · 2025-07-29
>
> We thank the reviewer for the time and the detailed review of our work.
>
> **How sensitive is the performance to the choice of covariate model**?
>
> Regarding your question regarding the choice of the covariate model, we perform a simulation study in which the covariate model cannot recover the data-generative model (see shifted tiled clustering, section 4). We show that the results are very similar to those of the EBMF, and thus using a completely misspecified covariate model has a rather limited impact on the cEBMF performance. See Figure 2 on the right panel.
>
> **What guarantees (if any) do you have about convergence or posterior consistency in this setting**?
>
> Similar to most fully factorized variational approximations, we can only guarantee that the fitting procedure will converge to a fixed point that is a local maximum of the ELBO, but we don't have posterior consistency results in this setting. Future work may focus on getting more precise on posterior consistency.
>
> **Is your method applicable in settings where covariates are high-dimensional or partially missing**?
>
> Our approach is applicable when the covariates are high-dimensional. See our sparsity-inducing simulations, where we have 1000 covariates for factors of length 200, and among the 1000 covariates, only three are actually affecting the factor sparsity. We show that cEBMF outperforms the other methods in this setting.
>
> Regarding missing values, this depends completely on the covariate model selected by the user; some have "built-in" missing value routines. In our paper, we use neural nets that modulate the prior weight of different mixture components.  However, it is possible to use XGBoost instead of a neural network, and XGBoost has some missing data routines already built-in. In this sense, cEBMF can be viewed as a "wrapper" that can benefit from the covariate model perks.
>
> We hope that we have answered your questions and that you may consider increasing our rating.

---

> > ### Comment · Reviewer_NLUp · 2025-08-03
> >
> > Thank you for the response and clarifications. In the light of the comments, I deem my original assessment appropriate.

---

### Note · Authors · 2025-08-13

Below we summarize some of the main points brought up by reviewers, and explain that these points have been at least partially addressed by our paper in theory and experiment:

**Novelty and Scope:**

cEBMF significantly extends Empirical Bayes Matrix Factorization (EBMF) by introducing covariate-modulated, adaptive priors, generalizing numerous approaches (SpatialPCA, CMF, MFAI), and supporting arbitrary side information (images, graphs, text) via modular integration of neural network/prior models. The framework provides broad applicability, e.g., genomics, recommender systems, and spatial data.

**Empirical Evaluation:**

Our experiments benchmark cEBMF against other recent covariate moderated matrix factorization (MFAI, Spatial PCA), machine learning approaches such as VAE/cVAE, and multiple baseline methods (including EBMF). Our benchmarks include ablation studies, tests on simulated and application on real data, including a recent spatial transcriptomics application. Overall, cEBMF shows excellent performance throughout our experiments, outperforming other methods. Further, we show in ablation studies that cEBMF performs well even when covariates or prior families are misspecified (see simulation, Figure 2), and scales markedly better than covariate-dependent alternatives ( MFAI, Spatial PCA).

**Robustness and Flexibility:**

Multiple reviewers inquired about robustness to covariate and prior mis-specification. We highlight (see Section 4, Appendix C) our extensive simulations showing cEBMF matches as similar performance as EBMF when covariates are uninformative and when using a grossly misspecified prior, cEBMF performance doesn't deteriorate much. We also show that cEBMF handles high-dimensional covariates well and can deal with missing data.

**Automatic Selection of K:**

Our appendix details how cEBMF can automatically determine the number of components in the factorization via empirical Bayes shrinkage (Bishop 1999; Wang and Stephens 2021), hence allowing automatic estimation of the matrix rank.

**Conclusion:**

We believe that cEBMF is suitable for publication in Neurips 2025 as it is a broadly useful method across application domains (e.g., genomics, recommender systems, spatial data), and it connects in a principled way the classical matrix factorization problem with modern machine learning approaches. Overall, cEBMF is both a practical and robust method that represents a step forward for modern matrix factorization methods.

---

### Decision · Program_Chairs · 2025-09-17

**Decision:**

Accept (poster)

**Comment:**

This paper introduces cEBMF, an empirical Bayes matrix factorization framework that flexibly incorporates side information into priors. Reviewers agreed that the framework is novel, modular, and well-presented, with empirical results across simulations, MovieLens, and spatial transcriptomics, as well as robustness to prior or covariate misspecification.

The main point of contention comes from Reviewer KUPC, who questions the positioning of cEBMF relative to modern end-to-end deep models and notes that the current evaluation (e.g., MovieLens 100K with limited covariates) may not fully substantiate broad claims. Other reviewers raised technical questions about robustness and initialization, which were addressed in rebuttal, and overall leaned to acceptance.

On balance, I find the contribution substantive and broadly useful. It offers a flexible extension of empirical Bayes factorization that accommodates diverse side information. While additional comparisons to state-of-the-art deep baselines would further strengthen the work, the majority of reviewers support acceptance, and I concur.